# Position: Temporal Measurement Interval Determines Computational and Model Complexity in Single-Cell Perturbation Analysis

Alireza Jafari [1]   Heman Shakeri [2]   Hadi Daneshmand [1]

## Abstract

Single-cell perturbation analysis aims to predict how cellular states change after interventions such as drug treatments or genetic edits. A central difficulty is that pre- and post-perturbation measurements are typically observed as *unpaired* populations, so accurate prediction requires inferring a latent coupling and learning a transition map. In this position paper, we argue that the *measurement time gap* is the key experimental knob controlling both the computational tractability of coupling and the effective model complexity. We identify a critical time gap $\Delta$ that induces a phase transition, under biologically inspired conditions; for "measurement-time $< \Delta$", matching is polynomial-time tractable and the task reduces to supervised learning, whereas for "measurement-time $> \Delta$", recovering the matching is NP-hard in the worst case. The required conditions are restricted isometry of the initial states and temporal smoothness of the transition dynamics. We complement the theory with empirical evidence on synthetic and biological datasets showing a sharp regime change as the time gap increases. Furthermore, we demonstrate that a linear model can match or exceed the performance of higher-capacity neural approaches when our conditions hold.

## 1. Introduction

Predicting how distributions change under interventions is a fundamental problem in machine learning, arising in various domains, such as biology. In single-cell perturbation analysis, the goal is to predict how cells respond to interventions such as drug treatments or genetic edits (Dixit et al., 2016), enabling applications in drug discovery (Szalai & Veres, 2023) and the prediction of patient-specific responses to therapies (Sinha et al., 2022).

A common theme across existing approaches is the use of highly expressive models. Neural optimal transport methods (Bunne et al., 2023), generative models (Lotfollahi et al., 2019), and transformer-based methods (Adduri et al., 2025) have been proposed to map pre-perturbation distributions to post-perturbation distributions, achieving strong empirical performance across multiple benchmarks. These approaches implicitly treat model selection as a post hoc step: data are collected first, and increasingly powerful models are then applied to explain the observed cellular responses.

In this position paper, we investigate a critical yet underexplored factor in single-cell perturbation analysis: the timing of measurements. Experimental designs typically observe post-perturbation cellular states at fixed times after an intervention is applied. **We establish a time-driven phase transition in computational complexity: below a critical time, single-cell perturbation analysis reduces to a standard supervised learning task; beyond that, it becomes NP-hard. We derive an explicit bound characterizing this transition that motivates a principled experimental design criterion.** This time gap depends on the temporal smoothness of the transition dynamics and on the restricted isometry property (RIP) of the initial states, a concept that is well studied in compressed sensing (Candes & Tao, 2005).

Building on these insights, we experimentally demonstrate that time can significantly reduce the required model complexity. While state-of-the-art approaches rely on highly nonlinear neural networks, we show that a linear model outperforms or matches these methods on multiple benchmarks when our theoretical assumptions hold. In contrast, when the assumptions are violated (e.g., measuring after the phase transition), none of the existing methods perform reliably, posing a fundamental challenge in perturbation analysis.

---
[*]Equal contribution  [1]Department of Computer Science, University of Virginia, Charlottesville, VA, USA. [2]Department of Data Science, University of Virginia, Charlottesville, VA, USA. Correspondence to: Alireza Jafari <jrp5td@virginia.edu>, Heman Shakeri <hs9hd@virginia.edu>, Hadi Daneshmand <dhadi@virginia.edu>.

*Proceedings of the 43rd International Conference on Machine Learning*, Seoul, South Korea. PMLR 306, 2026. Copyright 2026 by the author(s).

**Conflict of Interest Disclosure.** The authors declare no financial conflicts of interest related to this work.

## 2. Related Work

We classify existing perturbation-prediction methods into three categories: (1) optimal transport-based approaches, (2) generative models, and (3) transformer-based methods.

**(1) Optimal Transport-Based Methods**  The foundational work of Schiebinger et al. (2019) established optimal transport (OT) as a theoretical framework for modeling single-cell dynamics by computing couplings between pre- and post-perturbation distributions. To predict post-perturbation states, Bunne et al. (2023) introduced CellOT, which learns parametric OT maps via input-convex neural networks. CellOT excels on benchmarks involving scRNA-seq and multiplexed imaging data for drug responses, outperforming earlier methods.

Subsequent advances include Wasserstein-1 formulations for enhanced speed and scalability (Chen et al., 2025), unbalanced OT for cell proliferation or death (Lubeck et al., 2022), conditional Monge maps for better generalization to unseen perturbations (Driessen et al., 2025), and variants incorporating entropic regularization or Schrödinger bridges for stochastic mappings (Chi et al., 2025).

**(2) Generative Models for Perturbation Prediction** Generative models offer a flexible alternative for capturing distributional shifts in perturbation datasets. An early influential method, scGen (Lotfollahi et al., 2019), utilizes variational autoencoders (VAEs) with latent-space vector arithmetic to reverse perturbation effects, reconstruct control states, and predict responses across cell types and species. More recently, PerturbNet (Yu et al., 2025) employs normalizing flows to map perturbation features directly to cell-state distributions, enabling predictions for unseen compounds or genetic interventions. By leveraging expressive density estimation, it surpasses prior approaches. Like OT methods, generative models also require nonlinear neural networks.

**(3) Transformer-Based Methods**  Theoretical results by Daneshmand (2024) prove that multi-layer attention in transformers can approximate OT solutions by emulating gradient descent on dual potentials. Motivated by this insight, transformer-based models such as scGPT (Cui et al., 2024) and State (Adduri et al., 2025) have been developed for cell perturbation analysis. By pretraining on large scRNA-seq corpora and fine-tuning for perturbation prediction, these models aim to generalize across biological contexts.

**Challenge.**  Prior methods across all three categories rely heavily on nonlinear neural networks (e.g., input-convex networks, VAEs, normalizing flows, or transformers) to transport pre- to post-perturbation distributions. In contrast, we show that linear models can suffice for perturbation prediction when post-perturbation measurements are collected within a critical time gap. Building on OT frameworks from category (1), we replace neural networks with linear maps. We first derive an explicit bound characterizing when coupling recovery—and thus learning the transport map—is possible. Using synthetic data, we then show that beyond this threshold, recovering the exact linear transport map becomes NP-hard in the worst case. We further demonstrate that the time-driven phase transition appears in biological time-course data, and a linear model can match or outperform nonlinear baselines across multiple benchmarks.

## 3. Problem Statement

After a perturbation, cellular states evolve over time. Let $x_1^{(0)}, \ldots, x_n^{(0)} \in \mathbb{R}^d$ be i.i.d. pre-perturbation states drawn from an unknown distribution, and let $x_1^{(t)}, \ldots, x_n^{(t)} \in \mathbb{R}^d$ be post-perturbation states measured at time $t$, but *without known pairing* to the pre-perturbation cells. We assume there exists an *unknown* map $F_t : \mathbb{R}^d \to \mathbb{R}^d$ such that $x_i^{(t)} = F_t(x_{\sigma(i)}^{(0)})$ for some unknown permutation $\sigma$ of $[n] := \{1, 2, \ldots, n\}$. Figure 1 visualizes this perturbation process.

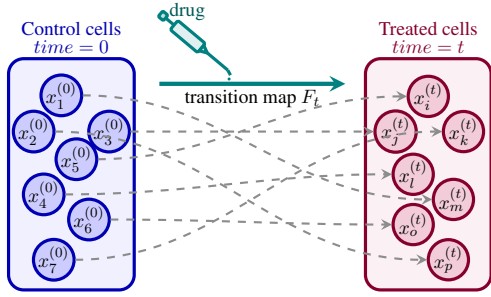

*Figure 1.* Drug effects on individual proteins/genomes are modeled by a transition function $F_t$, such that $x_i^{(t)} = F_t\left(x_{\sigma(i)}^{(0)}\right)$, where $\sigma$ is a permutation of $[n] = \{1, 2, \ldots, n\}$.

The goal is to learn a map $F_t$ that allows us to predict the post-perturbation state of a new cell (Bunne et al., 2023), namely, $x_{\text{test}}^{(t)} \approx F_t\left(x_{\text{test}}^{(0)}\right)$. By definition, $F_0$ is the identity map, i.e., $F_0(x) = x$. To formulate this compactly, we introduce the matrix $X^{(0)} \in \mathbb{R}^{d \times n}$ whose columns are $x_1^{(0)}, \ldots, x_n^{(0)}$, and similarly define the matrix $X^{(t)} \in \mathbb{R}^{d \times n}$ containing the columns $x_1^{(t)}, \ldots, x_n^{(t)}$. Cell perturbation analysis can thus be viewed as supervised learning with unknown correspondences between predictor and response variables.

Finding an exact cell-level matching $F_t$ between pre- and post-perturbation states may seem unachievable. However, $F_t$ can instead be interpreted as the optimal transport map between population distributions, whose existence is guaranteed under weak assumptions (Brenier, 1991). This connection between single-cell perturbation analysis and optimal transport theory was first established by Bunne et al. (2023).

# 4. Computational Complexity Analysis

We establish a time-dependent phase transition in computational complexity: when sampling time gaps are sufficiently fine, the perturbation dynamics remain near the identity, making linear models provably sufficient; under coarse sampling, the problem becomes computationally intractable.

## 4.1. Assumptions

Our results rely on biologically motivated insights and assumptions. In particular, we assume (i) *temporal smoothness* of the perturbation dynamics, and (ii) a *restricted isometry property* (RIP) for the pre-perturbation cell states. In our experiments, we explicitly demonstrate regimes where these assumptions hold and settings where they break down.

**Smooth perturbation dynamics.** Drug-induced perturbations act over time: a compound must engage its targets, trigger signaling cascades, and alter downstream transcriptional or translational programs before inducing measurable cellular responses.

**Assumption 4.1** (Lipschitz Transition)**.** *We assume that the transition map $F_t$ is Lipschitz continuous in time, i.e.,*

$$\sup_x \|F_t(x) - F_{t'}(x)\| \le L|t - t'|, \tag{1}$$

*for all $t, t'$, where $L > 0$ is a constant.*

We will experimentally show that violating the above condition leads to significantly harder perturbation prediction.

**Restricted isometry of initial states.** The second assumption governing our theory and experiments concerns the geometric structure of the pre-perturbation data. Specifically, we rely on a restricted isometry property, originally introduced and extensively studied in compressed sensing (Candes & Tao, 2005). We recall the definition below.

**Definition 4.2** (($k, \delta$)-Restricted Isometry Property (RIP) (Candes & Tao, 2005))**.** Let $A \in \mathbb{R}^{p \times q}$ and $k \in \mathbb{N}$. We say that $A$ satisfies the $(k, \delta)$-RIP if $\delta$ is the smallest constant such that

$$(1 - \delta)\|v\|_2^2 \le \|Av\|_2^2 \le (1 + \delta)\|v\|_2^2 \tag{2}$$

holds for all $k$-sparse vectors $v \in \mathbb{R}^q$.

Our analysis assumes that the data matrix $X^{(0)}$, whose columns correspond to pre-perturbation cell states, satisfies an RIP condition. To build intuition, we provide examples of matrices that do and do not satisfy this condition.

**Example 4.3** (Failure of RIP under strong correlations)**.** *Matrices with duplicated or highly correlated columns fail the $(k > 1, \delta)$-RIP. Indeed, if two columns of $X$ are identical, then the vector $u = e_i - e_j$ is 2-sparse but satisfies $Xu = 0$, violating the lower bound in (2).*

RIP conditions are well studied in compressed sensing (Candes & Tao, 2005; Unnikrishnan et al., 2018; Candes & Wakin, 2008). The following classical result illustrates a setting in which RIP holds with high probability.

**Example 4.4** (Gaussian ensemble)**.** *If the entries of $X \in \mathbb{R}^{d \times n}$ are drawn i.i.d. from a zero-mean Gaussian distribution with variance $1/d$, then $X$ satisfies the $(2, \sqrt{2} - 1)$-RIP with high probability, provided that $d \ge c \log(n)$ for an absolute constant $c$ (Candes & Wakin, 2008; Baraniuk et al., 2008; Rudelson & Vershynin, 2008; Candes & Tao, 2006).*

## 4.2. Nyquist Rate for Perturbation Analysis

The problem presents two fundamental challenges: (1) the samples observed before and after perturbation are unpaired, and (2) the transition map $F_t$ is unknown. If each post-perturbation sample $x_i^{(t)}$ could be matched to its corresponding pre-perturbation sample $x_{\sigma(i)}^{(0)}$, the task would reduce to standard supervised learning, where the goal is to estimate the transition function $F_t$. However, jointly recovering both the matching and the transition map is substantially more challenging than supervised learning alone.

The next theorem establishes a phase transition in the computational complexity of permutation recovery as a function of the perturbation time $t$.

**Theorem 4.5.** *Define the measurement time gap*

$$\Delta \coloneqq \sqrt{\frac{1 - \delta}{2nL^2}}.$$

*Then the problem of recovering the permutation $\sigma$ exhibits a phase transition as a function of time $t$:*

- ***Trackable regime** ($t < \Delta$). The permutation $\sigma$ can be recovered in polynomial time as long as $X^{(0)}$ satisfies the $(2, \delta)$-RIP condition and the transition map is Lipschitz continuous in time (Assumption 4.1).*

- ***Untrackable regime** ($t > \Delta$). Recovering $\sigma$, and consequently the transition map $F_t$, is NP-hard even when $F_t$ is a linear function.*

*The proof is given in Appendix A.*

Phase transitions with respect to time have been extensively studied in signal processing. In particular, the Nyquist rate characterizes a critical downsampling threshold below which signal recovery is fundamentally impaired by aliasing (Shannon, 1949). In contrast, despite the central role of time in biological experiments, its impact on the computational and statistical complexity of inference has remained largely unexplored. In this work, the sampling time gap $\Delta$ plays an analogous role and can be interpreted as a Nyquist-like rate for cell perturbation analysis.

The theorem reveals that the trackable time gap is jointly governed by the geometric properties of the problem, where

smoother transition dynamics (smaller Lipschitz constants) and robust initial RIP conditions directly extend the trackable time gap.

In the previous theorem, the critical time gap $\Delta$ shrinks as $n$ increases. To avoid this vanishing behavior, we derive a sharper bound in the special case where the transition map is a linear function (see Theorem B.1 in the appendix):

$$\Delta \;:=\; \frac{\sqrt{1-\delta}}{\sqrt{2}\,L\,\|X^{(0)}\|_F}.$$

Notably, this $\Delta$ does *not* scale with the number of samples $n$. We conjecture that a similar removal of the $n$-dependence is possible for more general (possibly nonlinear) transition maps. However, the dependence on the Lipschitz constant $L$ and the RIP constant $\delta$ is natural and intuitively justified.

## 5. Model Complexity Analysis

So far, we have analyzed how temporal resolution affects *computational* complexity. We now empirically show that time also governs *model* complexity. In particular, we study the problem of estimating a linear transition map of the form

$$F_t(x) = W_\star^{(t)}x, \tag{3}$$

where $W_\star^{(t)}$ is unknown. We first present an algorithm for estimating $F_t$, and then evaluate its performance for both small and large $t$ on synthetic and biological data. Our experiments reveal a sharp phase transition in performance as time increases, consistent with the prediction of Theorem 4.5. Moreover, results on a real-world dataset show that a linear model can perform well over short time intervals, a finding we further support in subsequent sections.

### 5.1. Linear Alternating Optimal Transport (LAOT): a minimal solver for linear transition

The fundamental difficulty in solving (3) is that the data are unpaired. As defined in Section 3, we observe control cells $X^{(0)} \in \mathbb{R}^{d \times n}$ and treated cells $X^{(t)} \in \mathbb{R}^{d \times n}$. Concretely, there exists an (unknown) permutation $\sigma : [n] \to [n]$ such that

$$x_i^{(t)} \;=\; W_\star^{(t)} x_{\sigma(i)}^{(0)} \qquad (i = 1, \ldots, n). \tag{4}$$

Let $\Pi_\star \in \Gamma$ be the permutation matrix induced by $\sigma$, so that $(X^{(0)}\Pi_\star)_{:,i} = X^{(0)}_{:,\sigma(i)}$, and $\Gamma$ denotes the Birkhoff polytope (the set of doubly stochastic matrices). Estimating the transition therefore amounts to recovering $(\Pi_\star, W_\star^{(t)})$ by least squares:

$$\min_{\Pi \in \Gamma,\, W^{(t)} \in \mathbb{R}^{d \times d}} \;\big\|X^{(t)} - W^{(t)}X^{(0)}\Pi\big\|_F^2. \tag{5}$$

The objective in (5) is not jointly convex in $(\Pi, W^{(t)})$. However, each block subproblem is polynomial-time solvable, motivating a natural alternating-minimization Algorithm 1, called Linear Alternating Optimal Transport (LAOT).

**Permutation update.** Fixing a candidate linear map $W^{(t)}$, the remaining ambiguity is the unknown matching between columns of $X^{(0)}$ and columns of $X^{(t)}$. Expanding the objective shows that the dependence on $\Pi$ is linear:

$$\|X^{(t)} - W^{(t)}X^{(0)}\Pi\|_F^2 = \|X^{(t)}\|_F^2 + \|W^{(t)}X^{(0)}\|_F^2 \\ -2\langle X^{(t)}, W^{(t)}X^{(0)}\Pi\rangle, \tag{6}$$

and $\|W^{(t)}X^{(0)}\|_F^2$ is invariant to $\Pi$ because $\Pi$ only permutes columns. Consequently, for fixed $W$, estimating $\Pi$ is equivalent to the linear assignment problem

$$\min_{\Pi \in \Gamma} \sum_{i,j} \Pi_{ij} \big\|(X^{(t)})_{:,i} - (W^{(t)}X^{(0)})_{:,j}\big\|_2^2. \tag{7}$$

This is the classical assignment formulation and admits efficient polynomial-time solvers. Moreover, since the objective in (7) is linear in $\Pi$ over the doubly stochastic polytope $\Gamma$, invoking the fundamental theorem of linear programming implies the minimizer is attained at an extreme point of $\Gamma$, known as permutation matrices (Brockett & Wong, 1991).

**Linear map update.** Once a matching $\Pi$ is fixed, the learning problem reduces to ordinary supervised estimation of the linear transition map: we have paired training data $(x_{\sigma(i)}^{(0)}, x_i^{(t)})$ encoded as $(X^{(0)}\Pi, X^{(t)})$, and $W^{(t)}$ is obtained by multivariate least squares,

$$W^{(t)} \in \arg\min_{W^{(t)} \in \mathbb{R}^{d \times d}} \|X^{(t)} - W^{(t)}X^{(0)}\Pi\|_F^2, \tag{8}$$

which is convex and admits a closed-form solution via the normal equations.

These two tractable updates together yield the following minimal procedure. We emphasize LAOT is not introduced for novelty, but as a minimal polynomial-time mechanism that isolates the source of computational hardness.

---

**Algorithm 1** Linear Alternating Optimal Transport (LAOT)

---

1: **Input:** Unpaired control samples $X^{(0)} \in \mathbb{R}^{d \times n}$ and treated samples $X^{(t)} \in \mathbb{R}^{d \times n}$; iterations $K$.
2: Initialize $W^{(t,0)} \leftarrow I_d$.
3: **for** $k = 1, 2, \ldots, K$ **do**
4:     Optimizing $\Pi$:

$$\Pi^{(k)} \in \arg\min_{\Pi \in \Gamma}\|X^{(t)} - W^{(t,k-1)}X^{(0)}\Pi\|_F^2,$$

    which is equivalent to solving (7).
5:     Optimizing $W^{(t)}$:

$$W^{(t,k)} \in \arg\min_{W \in \mathbb{R}^{d \times d}} \|X^{(t)} - W X^{(0)}\Pi^{(k)}\|_F^2.$$

6: **end for**
7: **Output:** $W^{(t,K)}$ and predictor $\widehat{x}_{\text{new}}^{(t)} = W^{(t,K)}x_{\text{new}}^{(0)}$.

---

## 5.2. Empirical validation of the time-driven phase transition

Theorem 4.5 establishes a *phase transition in time*: when the delay between pre- and post-perturbation measurements exceeds a critical threshold, recovering the matching becomes NP-hard. We empirically validate this time-driven transition using synthetic data and real single-cell data.

### 5.2.1. EMPIRICAL VALIDATION ON SYNTHETIC DATA

We validate the phase-transition mechanism on controlled synthetic data constructed to satisfy our standing assumptions: (i) we generate $X^{(0)}$ from i.i.d. Gaussian samples so that it obeys a $(2, \delta)$-RIP (Example 4.4), and (ii) we use a time-indexed linear transition map $F_t$ that is Lipschitz continuous in $t$ as in (3). We generate $n = 100$ samples and form $X^{(0)} \in \mathbb{R}^{d \times n}$ with entries drawn i.i.d. from $\mathcal{N}(0, 1/d)$. We draw a ground-truth permutation $\Pi_\star \in \Gamma$ uniformly at random. For each time $t$, we instantiate the transition in (3) by choosing a smooth path $t \mapsto W_\star^{(t)}$ that starts at $I_d$ and deforms monotonically with $t$; concretely, we draw a random matrix $E \in \mathbb{R}^{d \times d}$ with i.i.d. $\mathrm{Unif}[0, 1]$, and set $W_\star(t) = I_d + t\,E$. We then generate the matched time-$t$ population as $X_{\mathrm{paired}}^{(t)} = W_\star^{(t)} X^{(0)}$, and destroy pairing by permuting columns to obtain the observed unpaired data $X^{(t)} = X_{\mathrm{paired}}^{(t)} \Pi_\star$. We run Algorithm 1 and report the permutation recovery rate, averaged over 20 independent trials.

Figure 2 shows a sharp two-regime behavior: for fine time gap $t$, LAOT recovers the matching with near-perfect probability, while beyond a critical time scale the recovery rate rapidly collapses. Increasing $d$ expands the trackable regime, consistent with Theorem 4.5: larger $d$ supports the required RIP condition, whereas for small $d$ the standard RIP assumption is unlikely to be satisfied as $d \geq c \log(n)$ is required for RIP (Candes & Wakin, 2008). Appendix D.1 shows nonlinear baselines also collapse beyond the trackable regime.

### 5.2.2. EMPIRICAL VALIDATION ON BIOLOGICAL DATA

We next validate the same time-driven transition on a real single-cell time-course dataset: the reprogramming dataset of Schiebinger et al. (2019), which provides scRNA-seq measurements across multiple time resolutions. This dataset is a canonical benchmark for transport-based analysis because it naturally presents *unpaired* populations at successive times, matching the learning setting in (5). We focus on the 2i perturbation condition and use measurements sampled every 12 hours. For each horizon $\tau \in \{12, 24, \ldots, 168\}$ hours, we form an unpaired training set by taking control cells from the first measurement and treated cells from the time point $\tau$ hours later. We fit LAOT on training cells and evaluate on $n = 500$ held-out test cells, using the top 50 highly variable genes. Prediction quality is measured using the squared Maximum Mean Discrepancy (MMD$^2$) with RBF kernel and $\gamma = 0.1$ between the predicted and observed post-perturbation samples, following standard practice in prior work (Bunne et al., 2023; Lotfollahi et al., 2019). Additional details on the evaluation metric are provided in Appendix C.2. This setup allows us to validate the theory's concrete prediction: when the measurement time gap is sufficiently small (below the theoretical threshold $\Delta$), a linear model is adequate, whereas increasing the gap rapidly makes accurate prediction substantially more difficult.

Figure 3 exhibits a clear regime change as the prediction horizon $\tau$ increases. At the finest temporal gap ($\tau = 12$h), LAOT achieves a near-zero test discrepancy, consistent with a *trackable* regime in which correspondence recovery is stable. However, increasing the gap leads to a rapid increase in error: by $\tau = 36$h error increases by more than triple relative to $\tau = 12$h. Beyond this point, the curve enters a high-error regime. This sharp jump mirrors the synthetic results and provides empirical validation of the phase-transition phenomenon predicted by Theorem 4.5.

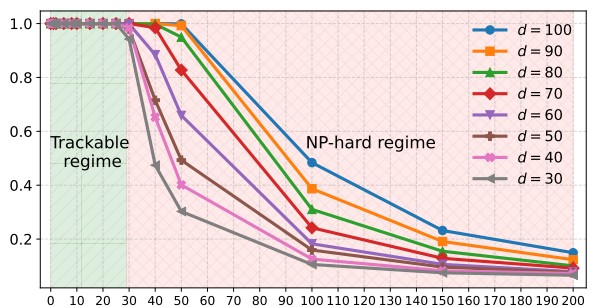

*Figure 2.* Phase transition in trackability on synthetic data. $y$-axis: average permutation recovery rate of LAOT over 20 trials. Data are generated with $n = 100$ i.i.d. Gaussian samples and a random permutation $\Pi_\star$. The ground-truth linear map is constructed as $W_\star(t) = I_d + t\,E$, so that $t$ is controlled exactly. $x$-axis: $t$.

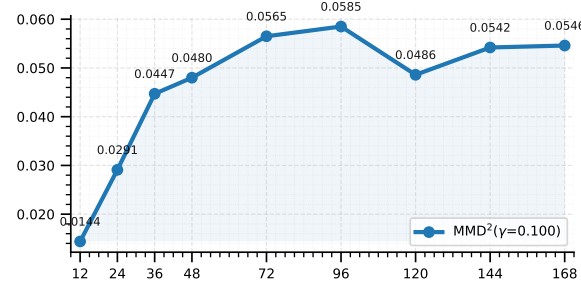

*Figure 3.* Empirical time-driven phase transition on the 2i time-course dataset. LAOT is trained on unpaired cells with controls from the first measurement and treated cells at horizon $\tau \in \{12, 24, \ldots, 168\}$ hours. $y$-axis: MMD$^2$ between LAOT predictions and observed post-perturbation populations on $n = 500$ held-out cells. $x$-axis: $\tau$.

# 6. Benchmarking under Trackability Conditions

Having established that correspondence recovery is stable when the measurement time gap $\Delta$ lies in the trackable regime of Theorem 4.5, we now verify the complementary implication: in this regime, once a matching is identified, learning the transition map reduces to supervised estimation, and a simple linear model suffices.

We benchmark across several standard single-cell perturbation datasets, as in prior work. Within each dataset, we evaluate a *within-context* setting, defined as a fixed biological context (same cell line and the same drug protocol), so the control and treated samples differ only through the drug-induced transition and drug responses evolve gradually over time, aligning with the theorem's smoothness condition. When replicates are available, we also evaluate cross-replicate generalization as a robustness check. Note that "within-context" is sometimes used more broadly to include unseen doses or closely related compounds; our benchmark datasets do not include these conditions.

Specifically, we validate on two protein-based perturbation datasets (AP-1 and 4i) and one scRNA-seq perturbation dataset (SciPlex3) to cover the two most common single-cell perturbation readouts: targeted protein panels (lower-dimensional marker measurements) and transcriptome-wide scRNA-seq (high-dimensional gene expression) (Gut et al., 2018; Comandante-Lou et al., 2022; Srivatsan et al., 2020). The measurement time gap of all datasets is relatively short ($t \leq 48$ hours).

## 6.1. Protein Perturbations Analysis

We evaluate perturbation prediction on two standard protein readouts (AP-1 and 4i) and compare the linear model LAOT with representative nonlinear baselines. Guided by biological considerations and supported by our empirical findings, we hypothesize that these benchmarks satisfy the assumptions of Theorem 4.5. This places them in a *trackable* regime. We begin by describing the benchmarks and baselines used in our experiments.

**Benchmark 1.** We use the AP-1 protein dataset (Comandante-Lou et al., 2022), which measures protein responses in melanoma cell lines under MAPK inhibition collected at a 48-hour post-treatment readout. We consider four commonly used melanoma cell lines: COLO858, WM902B, RVH421, and SKMEL19 (Comandante-Lou et al., 2022). Each cell is represented by a compact and interpretable JFE subset of $d = 10$ features (full feature list and biological motivation in Appendix C.1). Since data is collected at fine time resolution, we postulate that it lies within the trackable regime of Theorem 4.5. For each cell line, we train

on unpaired control/treated populations and evaluate distributional prediction on a held-out test split (80%/20%) using $\text{MMD}^2$ with an RBF kernel with a bandwidth chosen by the median of pairwise distance on the training set (see Appendix for other choices of bandwidths C.3). We report the $\text{MMD}^2$ on test data averaged over ten independent runs.

**Benchmark 2.** We also evaluate on the 4i protein-imaging perturbation dataset (Gut et al., 2018), which is used to evaluate one of the baselines (Bunne et al., 2023). This dataset profiles a 1:1 co-culture of two melanoma tumor cell lines (M130219 and M130429) under a panel of 34 drug treatments, with measurements taken after an 8-hour drug exposure. Each cell is represented by a $d = 48$ feature vector; the complete feature list is provided in Appendix C.1. We report representative results on Imatinib, Trametinib, and Dexamethasone. Notably, these span mechanistically distinct drug classes: Imatinib and Trametinib are kinase inhibitors acting on cytoplasmic signaling cascades, whereas Dexamethasone is a glucocorticoid that signals through nuclear receptor-mediated transcriptional regulation. Despite these mechanistic differences, the 8-hour measurement window is sufficiently short, keeping perturbations in the trackable regime of Theorem 4.5.

**Baselines.** We compare LAOT to representative nonlinear baselines discussed in Section 2. CellOT (Bunne et al., 2023) parameterizes the transport map with an input-convex neural network and jointly learns the OT coupling and the mapping; their default configuration has four hidden layers with 64 units each, yielding a parameter count that is orders of magnitude larger than LAOT. scGen (Lotfollahi et al., 2019) is a VAE-based generative model that learns a latent representation and predicts perturbation responses via latent-space arithmetic rather than explicit correspondence recovery, again relying on a comparatively large number of learnable parameters. We also modify Compact_CellOT, a reduced-capacity variant of CellOT (three hidden layers with 32 units each), used to isolate the role of model capacity and computational complexity.

**Results on protein perturbations.** Table 1 shows that LAOT achieves the lowest mean test $\text{MMD}^2$ on AP-1 and on two of the three drugs perturbation with 4i imaging, while remaining competitive on the remaining condition (cross-replicate results show the same trend in Appendix D.3). Notably, this accuracy is obtained with minimal capacity: LAOT learns only a single linear map (e.g., a $10 \times 10$ matrix on AP-1), initialized at the identity, with no additional parameters. These results align with the main message that *once the data collection protocol places the problem in the trackable regime: after recovering a reliable coupling, the remaining step is essentially supervised learning, for which a linear model can suffice.*

*Table 1.* Within-context protein perturbation prediction. Mean $\pm$ std $MMD^2$ between predicted and observed treated populations. AP-1: DMSO$\rightarrow$VEM at 48h. 4i: protein imaging after 8h exposure. C.CellOT stands for Compact_CellOT in this and all subsequent tables.

| Method | AP-1 dataset: Cell line | | | 4i dataset: Drug | | |
|---|---|---|---|---|---|---|
| | COLO858 | WM902B | SKMEL19 | Imatinib | Trametinib | Dexamethasone |
| CellOT | $0.0995\pm 0.1007$ | $0.0443\pm 0.0394$ | $0.1122\pm 0.0855$ | $0.0700\pm 0.0939$ | $0.0463\pm 0.0307$ | $0.0685\pm 0.0119$ |
| scGen | $0.0172\pm 0.0080$ | $0.1423\pm 0.0293$ | $0.0323\pm 0.0218$ | $0.0330\pm 0.0034$ | $0.0098\pm 0.0020$ | $0.0160\pm 0.0022$ |
| C.CellOT | $0.0019\pm 0.0007$ | $0.0015\pm 0.0006$ | $0.0016\pm 0.0004$ | $0.0079\pm 0.0035$ | $\mathbf{0.0076}\pm 0.0016$ | $0.0075\pm 0.0020$ |
| LAOT | $\mathbf{0.0006}\pm 0.0000$ | $\mathbf{0.0007}\pm 0.0000$ | $\mathbf{0.0011}\pm 0.0000$ | $\mathbf{0.0063}\pm 0.0000$ | $0.0080\pm 0.0000$ | $\mathbf{0.0071}\pm 0.0000$ |

**Challenge of nonlinear baselines.** Table 1 shows that representative nonlinear baselines such as CellOT and scGen underperform on these within-context, short-gap protein benchmarks. Theorem 4.5 suggests why: it distinguishes a *trackable* regime from an *untrackable* regime in which recovering a latent correspondence becomes NP-hard even when the underlying transition is linear. These nonlinear baselines, without explicit regime awareness, attempt to attack a computationally hard objective with a large, highly nonconvex parameterization, preventing the model from reaching stable optimization even in the tractable regime. In particular, both CellOT and scGen introduce tens of thousands of learnable parameters, while each drug–cell-line setting provides only on the order of thousands of cells. In their setting, optimization can become the bottleneck: the learning dynamics can be dominated by the difficulty of navigating a nonconvex landscape. In contrast, we modify Compact_CellOT for further evidence for this interpretation. By reducing the number of learnable parameters in the default CellOT architecture, Compact_CellOT improves optimization stability. In Table 1, it sometimes comes close to LAOT when its training converges well. Figure 4 supports this interpretation on optimization stability: CellOT exhibits highly non-monotone training with substantial $MMD^2$ oscillations; reducing capacity (Compact_CellOT) stabilizes the trajectory but still produces long plateaus and abrupt drops; in contrast, LAOT exhibits more stable convergence.

## 6.2. scRNA-seq Perturbations Analysis

We next turn to a transcriptome-wide perturbation readout to test whether the same conclusion extends beyond targeted protein panels. In particular, we evaluate within-context drug-response prediction on a higher-dimensional multiplexed scRNA-seq benchmark, which is often viewed as a setting where nonlinear representation learning is essential.

**Benchmark 3.** We evaluate on SciPlex3, a high-dimensional multiplexed scRNA-seq perturbation dataset measuring transcriptional responses to drugs, taken after 24-hour (Srivatsan et al., 2020). In this experiment, we use the standard CellOT benchmark protocol: models are trained on 1,000 genes, predictions are produced in the full 1,000-dimensional space, and evaluation is computed using $MMD^2$ on the top 100 highly variable genes, as in (Bunne et al., 2023). This evaluation choice is also practically motivated, since kernel-based distances such as MMD can become less sensitive and more bandwidth-dependent in very high dimensions. As discussed in Example 4.4, larger feature dimensions make the RIP condition easier to satisfy under Gaussian features, suggesting that this benchmark more closely satisfies the RIP required for trackability.

**Results on scRNA-seq perturbations.** Table 2 summarizes results on three commonly benchmarked perturbations: Trametinib, Givinostat, and Abexinostat. Despite SciPlex3 being a transcriptome-wide scRNA-seq benchmark with sub-

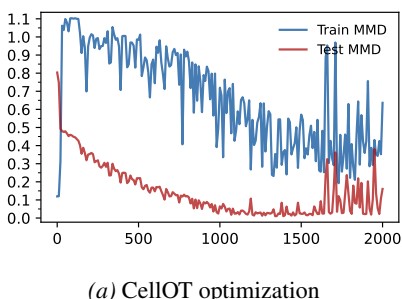

*(a)* CellOT optimization

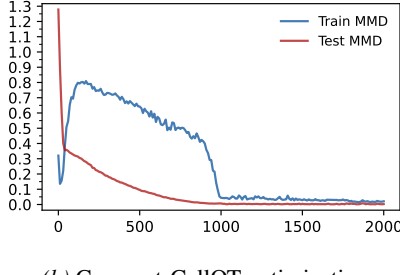

*(b)* Compact_CellOT optimization

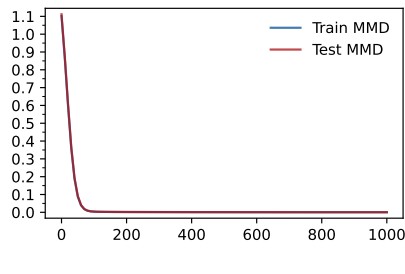

*(c)* LAOT optimization

*Figure 4.* Optimization stability in the within-context AP-1 task (COLO858, DMSO$\rightarrow$VEM). Train and test $MMD^2$ traces across iterations using the same split and $MMD^2$ setup. CellOT exhibits highly non-monotone training dynamics, whereas capacity reduction stabilizes training but can introduce long plateaus. LAOT converges rapidly with near-coincident train/test curves.

stantially larger feature dimension, a modality often viewed as requiring nonlinear representation learning, LAOT attains the lowest MMD$^2$ in all three cases, outperforming the nonlinear baselines. These outcomes are consistent with the observations on the protein-based perturbation datasets AP-1 and 4i (more experiments in Appendix D.4).

*Table 2.* Within-context scRNA-seq perturbation prediction. Mean $\pm$ std MMD$^2$ for SciPlex3, measurement after 24h drug exposure.

| Method | SciPlex3 dataset: Drug | | |
|---|---|---|---|
| | **Trametinib** | **Givinostat** | **Abexinostat** |
| CellOT | $0.0078\pm$ 0.0020 | $0.0117\pm$ 0.0029 | $0.0129\pm$ 0.0063 |
| scGen | $0.0059\pm$ 0.0014 | $0.0083\pm$ 0.0007 | $0.0091\pm$ 0.0012 |
| C.CellOT | $0.0048\pm$ 0.0010 | $0.0079\pm$ 0.0011 | $0.0074\pm$ 0.0019 |
| LAOT | $\mathbf{0.0040}\pm$ 0.0000 | $\mathbf{0.0033}\pm$ 0.0000 | $\mathbf{0.0038}\pm$ 0.0000 |

These results suggest that high feature dimension is not inherently detrimental; increasing $d$ can *help* by making RIP non-degeneracy conditions easier to satisfy (see Example 4.4), expanding the trackable regime.

## 7. Alternative Views

Our position reframes single-cell perturbation prediction as a regime-dependent problem in which both the required model expressivity and the feasibility of correspondence recovery are governed by the measurement time gap. Below, we summarize two complementary perspectives.

**Model-centric view: expressive architectures.** One line of work treats single-cell perturbation prediction as an inherently complex, nonlinear problem and responds by increasing model expressivity. Our results do not dispute this intuition; rather, they place it in a regime-dependent context. In particular, when measurements are taken before the defined time gap $\Delta$, the learning problem reduces to supervised regression once pairing is recovered, and a standard linear model can suffice. Conversely, when measurements are taken at time gaps larger than $\Delta$, computational complexity becomes the primary bottleneck—even when the transition is as simple as a linear map.

**Design-first: designing for learnability.** Another implicit assumption in many workflows is a sequential pipeline: choose an experiment, collect unpaired populations at a fixed time point, and then treat model selection as a post hoc step by applying powerful architectures. Our position argues for reversing this logic. Experimental design can *shape* the computational difficulty of the learning problem and directly influence the appropriate model class. When protocol choices keep the system within a trackable regime, the analysis admits simpler and more interpretable models. Hence, model design and experimental design should be treated as

mutually informative steps, rather than independent stages.

## 8. Discussion and Limitations

Our position motivates a model-informed data collection for single-cell perturbation analysis: experimental protocols should aim to collect measurements within the trackable regime predicted by Theorem 4.5. In principle, this requires measurement gaps below the critical threshold $\Delta$. While $\Delta$ is difficult to compute a priori because the Lipschitz and RIP constants are hard to estimate, it can be estimated *empirically* for each drug and cell-line. By running LAOT on a short pilot time-course over a few candidate measurement gaps, researchers can monitor when matching recovery and prediction error begin to deteriorate, thereby identifying a tractable temporal window. Biological priors, such as pathway activation times or protein/RNA half-lives, may further guide this search, though principled biological estimates of $\Delta$ remain future work.

A challenge observed under out-of-distribution generalization, where the biological context itself shifts and the assumptions behind Theorem 4.5 are no longer satisfied. We exemplify cross-cell-line prediction: training on some cell lines and evaluating on a held-out cell line; in this setting, the distribution shift is often substantial, breaking the assumption of smooth transition. This setting couples three hard problems simultaneously, recovering a latent coupling, estimating a context-dependent transition, and compensating for a substantial distribution shift. Consistent with this interpretation, Table 3 shows a clear performance degradation relative to within-cell-line prediction. In several cases, a method that performs well for one held-out cell line is far from best for another, indicating that neither LAOT nor the nonlinear baselines reliably resolve this cross-context shift. We therefore interpret this experiment not as contradicting the trackability result, but as clarifying its scope: when the biological context shifts substantially, the smooth-transition/Lipschitz assumption may no longer hold, moving the problem outside the trackable regime where no evaluated method is consistently successful. Future work could develop principled experimental and algorithmic criteria to *detect* and *enforce* trackable regimes in such settings.

*Table 3.* Cross–cell-line generalization in AP-1 proteins. Mean $\pm$ std MMD$^2$. Models are trained on a set of cell lines and evaluated on a held-out cell line (more information in Appendix D.5).

| Method | AP-1 dataset: Held-out cell line | | |
|---|---|---|---|
| | **COLO858** | **WM902B** | **SKMEL19** |
| CellOT | $0.158\pm$ 0.055 | $0.172\pm$ 0.016 | $0.229\pm$ 0.159 |
| C.CellOT | $0.227\pm$ 0.027 | $\mathbf{0.120}\pm$ 0.025 | $0.217\pm$ 0.035 |
| scGen | $0.102\pm$ 0.040 | $0.311\pm$ 0.069 | $\mathbf{0.117}\pm$ 0.016 |
| LAOT | $\mathbf{0.067}\pm$ 0.000 | $0.171\pm$ 0.000 | $0.254\pm$ 0.000 |

A caveat is that verifying the RIP assumption in Theorem 4.5 is generally hard for real biological data: computing or certifying the relevant RIP constant for a given cell-state matrix is not practical. Nonetheless, RIP can be interpreted as a standard "well-spread geometry" condition from compressed sensing and holds with high probability for common random-design models, such as matrices with i.i.d. Gaussian entries, as described in Example 4.4.

A central clarification is that our results do not claim that biological perturbations are universally linear. Instead, our claim is that in the trackable regime, recovering a latent correspondence can become tractable, after which the remaining task reduces to supervised learning and may still require a nonlinear predictor. LAOT is therefore best viewed as a minimal solver that isolates the role of measurement timing and model complexity, not as a universal model for all perturbation responses. Rapid nonlinear shock responses, even over short time intervals, may violate the linear approximation and require nonlinear supervised models after matching. The fact that Compact_CellOT slightly outperforms LAOT in one trackable setting is consistent with this view, but adopting highly expressive nonlinear architectures is not always necessary when the experimental design places the task in a trackable regime.

Our current formulation focuses on the balanced setting, where the pre- and post-perturbation populations are matched by a permutation. This is a useful abstraction for studying the computational difficulty of latent correspondence, but real perturbations can also involve cell proliferation, cell death, or changes in population mass. Extending our analysis to unbalanced transportation is an important direction for future work.

## Acknowledgements

Heman Shakeri was partially supported by the University of Virginia Comprehensive Cancer Center. We acknowledge the support of the UVA Engineering Dean's startup fund for supporting Hadi Daneshmand and Alireza Jafari.

## Generative AI Disclosure

Generative AI tools were used to assist with manuscript editing, clarity improvements, general feedback, and parts of the implementation, including evaluation utilities, plotting scripts, documentation, and reproducibility instructions. All scientific claims, experimental results, final manuscript text, and released code were reviewed, edited, tested, and validated by the authors.

## Code Availability

To support reproducibility, we release the full implementation of our experiments, including LAOT, baseline configurations, synthetic phase-transition experiments, and scripts for reproducing the tables and figures in the main text and appendix. The code is publicly available in our GitHub repository[1].

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

## A. Proof of Theorem 4.5 (Nyquist Rate for Perturbation Analysis)

**Restated Theorem.** Suppose $X_0$ satisfies the $(2, \delta)$-RIP condition and the transition map is Lipschitz continuous in time, as stated in Assumption 4.1. Define the measurement rate

$$\Delta^2 := \frac{(1-\delta)}{2nL^2}.$$

Then the problem of recovering the permutation $\sigma$ exhibits a phase transition as a function of time $t$:

- **Trackable regime** $(t < \Delta)$. The permutation $\sigma$ can be recovered in polynomial time.

- **Untrackable regime** $(t > \Delta)$. Recovering $\sigma$, and consequently the transition map $F_t$, is NP-hard.

**Trackable regime** $(t < \Delta)$**: a polynomial-time recovery algorithm.** The proof is constructive: we provide a method to recover the permutation $\sigma$ and to match post-perturbation states to pre-perturbation ones. Let $\mathcal{S}$ denote the set of $n \times n$ doubly stochastic matrices. We define the following convex program:

$$\widehat{\Pi} = \arg\min_{\Pi \in \mathcal{S}} \|X^{(0)}\Pi - X^{(t)}\|_F^2. \tag{9}$$

Notably, the solution to the above problem is equivalent to the solution of the following linear program:

$$\arg\min_{\Pi \in \mathcal{S}} \|X^{(0)}\Pi - X^{(t)}\|_F^2 = \underbrace{\|X^{(0)}\|_F^2}_{=\|X^{(0)}\Pi\|_F^2} + 2 \arg\min_{\Pi \in \mathcal{S}} \langle X^{(0)}\Pi, X^{(t)} \rangle + \|X^{(t)}\|_F^2. \tag{10}$$

Since the above problem is a linear convex program, the fundamental theorem of linear programming implies that its solution lies at an extreme point of the set of doubly stochastic matrices. It is well known that the extreme points of this set are permutation matrices. Thus, $\widehat{\Pi}$ is a permutation matrix. Thus, $\widehat{\Pi}$ provides a matching between pre and post perturbation states. We establish sufficient conditions ensuring that this permutation matrix recovers $\sigma$.

Define $M \in \mathbb{R}^{d \times n}$ whose columns are $F_t(x_1^{(0)}), F_t(x_2^{(0)}), \ldots, F_t(x_n^{(0)})$. By definition, there exists a permutation matrix $\Pi_*$ such that

$$X^{(t)} = M\Pi_*. \tag{11}$$

We are now ready to state the contradiction argument below:

**Contradiction:** Assume that $\widehat{\Pi} \neq \Pi_*$. We show that this implies $t \geq \Delta$, which contradicts to $t < \Delta$.

The Lipschitz transition assumption 4.1 implies

$$\|M - X\|_F^2 = \sum_{i=1}^n \|x_i^{(0)} - F_t(x_i^{(0)})\|^2 \leq nL^2t^2, \tag{12}$$

We will establish the following lower and upper bounds:

$$2(1-\delta) \leq \|X\widehat{\Pi} - X\Pi_*\|_F^2 \leq 4nL^2t^2, \tag{13}$$

which implies

$$\Delta^2 = \frac{(1-\delta)}{2nL^2} \leq t^2, \tag{14}$$

contradicting the assumption $t < \Delta$. It remains to prove upperbound and lowerbound (13) hold.

**Upperbound in** (13). The key to establishing the upper bound is the optimality of $\widehat{\Pi}$:

$$\|X\widehat{\Pi} - X^{(t)}\|_F^2 = \|X\widehat{\Pi} - M\Pi_*\|_F^2 \tag{15}$$

$$\leq \|X\Pi_* - M\Pi_*\|_F^2 \tag{16}$$

$$\leq \|X - M\|_F^2 \tag{17}$$

$$\leq nL^2t^2. \tag{18}$$

Here, (16) holds because $\widehat{\Pi}$ is the optimal solution of (9), (17) follows since $\Pi_*$ is orthogonal, and (18) is obtained by substituting (12).

These results yield

$$\|X\widehat{\Pi} - X\Pi_*\|_F^2 = \|X\widehat{\Pi} - M\Pi_* + M\Pi_* - X\Pi_*\|_F^2 \tag{19}$$

$$\leq 2\|X\widehat{\Pi} - M\Pi_*\|_F^2 + 2\|M\Pi_* - X\Pi_*\|_F^2 \tag{20}$$

$$\overset{(18)}{\leq} 2nL^2t^2 + 2\|M\Pi_* - X\Pi_*\|_F^2 \tag{21}$$

$$\leq 2nL^2t^2 + 2\|M - X\|_F^2 \tag{22}$$

$$\overset{(12)}{\leq} 4nL^2t^2 \tag{23}$$

**Proving the lower bound in** (13). Since $\widehat{\Pi} \neq \Pi_*$, there exist indices $i \neq j$ such that

$$e_i\widehat{\Pi}\Pi_*^\top = e_j,$$

where $e_i$ denotes the $i$th standard basis vector in $\mathbb{R}^n$. Using this observation, we obtain

$$\|X\Pi_* - X\widehat{\Pi}\|_F^2 = \|X(I - \widehat{\Pi}\Pi_*^\top)\|_F^2 \tag{24}$$

$$\geq \|X(I - \widehat{\Pi}\Pi_*^\top)e_i\|_2^2 \tag{25}$$

$$= \|X(e_i - e_j)\|_2^2 \tag{26}$$

$$\geq 2(1 - \delta). \tag{27}$$

The steps above are justified as follows. Equation (24) follows from the orthogonality of $\Pi_*$. Equation (25) uses the identity $\|A\|_F^2 = \sum_{j=1}^n \|Ae_j\|_2^2$. Equation (26) follows since $(I - \widehat{\Pi}\Pi_*^\top)e_i = e_i - e_j$. Finally, (27) follows from the $(2, \delta(X))$-RIP assumption (Definition 4.2), noting that $e_i - e_j$ is 2-sparse with norm $\sqrt{2}$.

**Untrackable regime** ($t > \Delta$)**: NP-hardness in the worst case.** We show that even the *linear* case is NP-hard.

We reduce from the NP-hard problem studied by (Pananjady et al., 2016). They consider the model

$$y = \Pi Ax,$$

with $y \in \mathbb{R}^n$, $A \in \mathbb{R}^{n \times d}$, $\Pi$ an $n \times n$ permutation matrix, and $x \in \mathbb{R}^d$ an unknown vector, and show (via a reduction from PARTITION) that the decision problem

Given $(A, y)$, does there exist $(\Pi, x)$ such that $y = \Pi Ax$?

is NP-hard whenever $d > 1$ (Theorem 4 in (Pananjady et al., 2016)).

Moreover, in their reduction from PARTITION, they construct $y \in \mathbb{Z}^{2d+1}$ and $A \in \mathbb{Z}^{(2d+1) \times 2d}$ of the form

$$A = \begin{bmatrix} I_{2d} \\ \mathbf{1}_d^\top & -\mathbf{1}_d^\top \end{bmatrix},$$

so that $A$ has full column rank (hence $\ker(A) = \{0\}$), and they prove that $(A, y)$ is a YES-instance of the above problem if and only if the original PARTITION instance is a YES-instance.

Fix such an instance $(A, y)$. Construct an instance $(X^{(0)}, X^{(t)})$ of our linear-transition model by

$$X^{(0)} := A^\top \in \mathbb{R}^{d \times n}, \qquad X^{(t)} := \begin{bmatrix} y^\top \\ 0 \\ \vdots \\ 0 \end{bmatrix} \in \mathbb{R}^{d \times n},$$

i.e., the first row of $X^{(t)}$ is $y^\top$ and the remaining $d - 1$ rows are identically zero. This construction is polynomial in the size of $(A, y)$.

**Claim.** There exists $(\Pi, W)$ with $X^{(t)} = W X^{(0)} \Pi$ if and only if there exists $(\Pi', x)$ with $y = \Pi' A x$.

*($\Rightarrow$).* Suppose $X^{(t)} = W X^{(0)} \Pi$ holds. Taking transposes and using $X^{(0)} = A^\top$ gives

$$(X^{(t)})^\top = \Pi^\top A W^\top.$$

Let $\Pi' := \Pi^\top$ (still a permutation matrix) and write $W^\top = [w^{(1)} \cdots w^{(d)}]$. Since $(X^{(t)})^\top = [y \; 0 \; \cdots \; 0]$, we have for every $j \geq 2$,

$$0 = (X^{(t)})^\top_{(:,j)} = \Pi' A w^{(j)} \quad \Rightarrow \quad A w^{(j)} = 0.$$

Because $A$ has full column rank, $w^{(j)} = 0$ for all $j \geq 2$. Hence $w^{(1)} = x$ for some $x \in \mathbb{R}^d$ and

$$y = (X^{(t)})^\top_{(:,1)} = \Pi' A w^{(1)} = \Pi' A x,$$

so $(A, y)$ is a YES-instance of the Pananjady problem.

*($\Leftarrow$).* Conversely, suppose there exist $\Pi' \in \mathbb{R}^{n \times n}$ and $x \in \mathbb{R}^d$ with $y = \Pi' A x$. Define

$$W^\top := [x \; 0 \; \cdots \; 0] \in \mathbb{R}^{d \times d} \quad \text{and} \quad \Pi := (\Pi')^\top.$$

Then

$$(X^{(t)})^\top = [y \; 0 \; \cdots \; 0] = \Pi' A W^\top = \Pi^\top A W^\top,$$

and transposing yields $X^{(t)} = W X^{(0)} \Pi$.

Therefore, deciding whether there exists $(\Pi, W)$ such that $X^{(t)} = W X^{(0)} \Pi$ is NP-hard for $d > 1$. Since this linear-transition case is a restriction of our general correspondence recovery problem, permutation recovery is NP-hard in the worst case in the untrackable regime.

## B. Complexity for Linear Transition Functions

We state and prove a specialization of our general phase-transition Theorem 4.5 to a *near-identity* transition. In this setting, the post-perturbation matrix satisfies $X^{(t)} = W^{(t)} X^{(0)} \Pi_\star$ where $W^{(0)} = I_d$ and $\|W^{(t)} - I_d\|_2 \leq Lt$, so the deviation from identity grows at most linearly with the measurement time gap $t$.

**Theorem B.1** (Permutation recovery under a near-identity transition). *Let $X^{(0)} \in \mathbb{R}^{d \times n}$ satisfy the $(2, \delta)$-RIP. Suppose the transition is linear with $X^{(t)} = W^{(t)} X^{(0)} \Pi_\star$, where $\Pi_\star \in \mathbb{R}^{n \times n}$ is a permutation matrix, $W^{(0)} = I_d$, and $\|W^{(t)} - I_d\|_2 \leq Lt$. Consider the first permutation update of LAOT initialized at $W^{(0)} = I_d$:*

$$\Pi_0 \in \arg\min_{\Pi \in \Gamma} \|X^{(t)} - X^{(0)} \Pi\|_F^2.$$

*If*

$$t < \Delta := \frac{\sqrt{1 - \delta}}{\sqrt{2} \, L \, \|X^{(0)}\|_F},$$

*then $\Pi_0 = \Pi_\star$; hence the correspondence is recovered in polynomial time (by solving the above assignment).*

### B.1. Proof of Theorem B.1 (Permutation Recovery Under a Linear Transition)

**Notations and shapes.** Let $X^{(0)} \in \mathbb{R}^{d \times n}$ be the pre-perturbation matrix whose $i$th column is $x_i^{(0)} \in \mathbb{R}^d$. Let $X^{(t)} \in \mathbb{R}^{d \times n}$ be the post-perturbation matrix at time $t$. In the linear setting, assume

$$X^{(t)} = W^{(t)} X^{(0)} \Pi_\star, \qquad W^{(t)} \in \mathbb{R}^{d \times d}, \quad \Pi_\star \in \mathbb{R}^{n \times n} \text{ (permutation matrix).} \tag{28}$$

Let $\Gamma$ denote the set of $n \times n$ permutation matrices.

**Assumptions.**

- $(2, \delta)$-**RIP for $X^{(0)}$.** For every 2-sparse $v \in \mathbb{R}^n$,

$$(1 - \delta)\|v\|_2^2 \leq \|X^{(0)} v\|_2^2 \leq (1 + \delta)\|v\|_2^2. \tag{29}$$

- **Lipschitz-in-time (near-identity) for the linear map.** Assume $W^{(0)} = I_d$ and

$$\|W^{(t)} - I_d\|_2 \leq Lt. \tag{30}$$

  Consequently,

$$\|(W^{(t)} - I_d)X^{(0)}\|_F \leq \|W^{(t)} - I_d\|_2 \, \|X^{(0)}\|_F \leq Lt \, \|X^{(0)}\|_F, \qquad \|(W^{(t)} - I_d)X^{(0)}\|_F^2 \leq L^2 t^2 \|X^{(0)}\|_F^2. \tag{31}$$

**Algorithmic step (first permutation update).** Under the initialization $W^{(0)} = I_d$, the first permutation update in LAOT returns

$$\Pi_0 \in \arg\min_{\Pi \in \Gamma} \|X^{(t)} - X^{(0)} \Pi\|_F^2 = \arg\min_{\Pi \in \Gamma} \|W^{(t)} X^{(0)} \Pi_\star - X^{(0)} \Pi\|_F^2. \tag{32}$$

Define $\widetilde{\Pi}_0 := \Pi_0 \Pi_\star^\top \in \Gamma$. Since $\Pi_\star$ is orthogonal and the Frobenius norm is invariant under right-multiplication by orthogonal matrices,

$$\|W^{(t)} X^{(0)} \Pi_\star - X^{(0)} \Pi_0\|_F = \|(W^{(t)} X^{(0)} \Pi_\star - X^{(0)} \Pi_0) \Pi_\star^\top\|_F = \|W^{(t)} X^{(0)} - X^{(0)} \widetilde{\Pi}_0\|_F. \tag{33}$$

Thus, $\widetilde{\Pi}_0$ minimizes $\|W^{(t)} X^{(0)} - X^{(0)} \widetilde{\Pi}\|_F$ over $\widetilde{\Pi} \in \Gamma$, and the ground truth corresponds to $\widetilde{\Pi} = I_n$.

**Proof Idea: Contradiction.** Assume for contradiction that $\Pi_0 \neq \Pi_\star$, equivalently $\widetilde{\Pi}_0 \neq I_n$. Under these conditions, we will show that $t \geq \Delta$ holds with contradicts to $t < \Delta$.

By the triangle inequality,

$$\|X^{(0)} - X^{(0)} \widetilde{\Pi}_0\|_F \leq \|X^{(0)} - W^{(t)} X^{(0)}\|_F + \|W^{(t)} X^{(0)} - X^{(0)} \widetilde{\Pi}_0\|_F$$
$$= \|(W^{(t)} - I_d)X^{(0)}\|_F + \|W^{(t)} X^{(0)} - X^{(0)} \widetilde{\Pi}_0\|_F. \tag{34}$$

By optimality of $\widetilde{\Pi}_0$ (using (33)), we have

$$\|W^{(t)}X^{(0)} - X^{(0)}\widetilde{\Pi}_0\|_F \le \|W^{(t)}X^{(0)} - X^{(0)}I_n\|_F = \|(W^{(t)} - I_d)X^{(0)}\|_F. \tag{35}$$

Substituting into (34) yields

$$\|X^{(0)} - X^{(0)}\widetilde{\Pi}_0\|_F \le 2\|(W^{(t)} - I_d)X^{(0)}\|_F. \tag{36}$$

Squaring and applying (31) gives

$$\|X^{(0)} - X^{(0)}\widetilde{\Pi}_0\|_F^2 \le 4\|(W^{(t)} - I_d)X^{(0)}\|_F^2 \le 4L^2t^2\|X^{(0)}\|_F^2. \tag{37}$$

Since $\widetilde{\Pi}_0 \ne I_n$, there exist indices $i \ne j$ such that $\widetilde{\Pi}_0 e_i = e_j$, where $e_i$ is the $i$th standard basis vector in $\mathbb{R}^n$. Using $\|A\|_F^2 = \sum_{k=1}^n \|Ae_k\|_2^2$ for $A \in \mathbb{R}^{d \times n}$,

$$
\begin{aligned}
\|X^{(0)} - X^{(0)}\widetilde{\Pi}_0\|_F^2 &= \|X^{(0)}(I_n - \widetilde{\Pi}_0)\|_F^2 \\
&\ge \|X^{(0)}(I_n - \widetilde{\Pi}_0)e_i\|_2^2 \\
&= \|X^{(0)}(e_i - e_j)\|_2^2 \\
&\ge (1-\delta)\|e_i - e_j\|_2^2 \qquad \text{(by } (2,\delta)\text{-RIP (29))} \\
&= 2(1-\delta).
\end{aligned} \tag{38}
$$

Combining (37) and (38) yields

$$2(1-\delta) \le 4L^2t^2\|X^{(0)}\|_F^2 \quad \Longrightarrow \quad t^2 \ge \frac{1-\delta}{2L^2\|X^{(0)}\|_F^2}. \tag{39}$$

Therefore, if

$$t < \Delta \quad \text{where} \quad \Delta = \frac{\sqrt{1-\delta}}{\sqrt{2}\,L\,\|X^{(0)}\|_F}, \tag{40}$$

the assumption $\widetilde{\Pi}_0 \ne I_n$ is impossible. Hence $\widetilde{\Pi}_0 = I_n$, which implies $\Pi_0 = \Pi_\star$, completing the proof.

# C. Experimental Setting

## C.1. Datasets and experimental protocols

**Why these biological benchmarks?** We choose a set of widely used single-cell perturbation benchmarks that are *complementary along the axes that matter for perturbation prediction in practice*: (i) *readout modality* (targeted protein panels, multiplexed imaging, scRNA-seq), (ii) *state dimensionality and inductive bias* (compact mechanistic marker sets vs. transcriptome-wide responses), (iii) *biological heterogeneity* (multiple melanoma cell lines; mixed co-cultures), and (iv) *protocol realism* (standard drug-response setups with unpaired populations, and—when available—replicates). This benchmark suite is designed to ensure that any empirical claim we make is not an artifact of a single modality, a single feature dimension, or a single biological context.

Across datasets, our primary evaluation is *within-context* (fixed biological context: same cell line and the same perturbation protocol), which isolates the perturbation-induced change from larger sources of distribution shift (e.g., across cell lines). When replicates are available, we additionally report *cross-replicate* generalization as a robustness check against measurement noise, batch effects, and replicate-specific artifacts.

**Benchmark 1: AP-1 protein perturbations (targeted panel).** We use the AP-1 protein dataset (Comandante-Lou et al., 2022), which measures MAPK-regulated protein responses in melanoma cell lines under MAPK inhibition, with DMSO as control and vemurafenib (VEM) as treatment, collected at a 48-hour post-treatment readout. *Motivation.* AP-1 is a *mechanistically grounded, low-dimensional* setting: the readout concentrates on a signaling/transcription-factor network (AP-1 and associated markers) whose components are interpretable and directly tied to MAPK inhibitor response. This makes AP-1 a good stress-test for whether a model can exploit *structure* rather than brute-force capacity. Multiple melanoma cell lines (COLO858, WM902B, RVH421, SKMEL19) further let us separate "works in one context" from "robust across closely related contexts," without moving into full out-of-context generalization. For each cell line, we train on *unpaired* control/treated populations and evaluate distributional prediction on a held-out test split (80%/20%), reporting mean±std $\text{MMD}^2$ over ten independent runs. *Data availability:* study page and Supplementary Data S4

For the AP-1 benchmark, we use a compact $d = 10$ representation ("JFE": JUN/FOS family + ERK) consisting of *p-cFOS*, *p-cJUN*, *cFOS*, *cJUN*, *FRA2*, *JUNB*, *JUND*, *FRA1*, *p-FRA1*, and *p-ERK*. The first nine variables summarize the core AP-1 transcription-factor module (Jun/Fos/Fra) together with phosphorylation-based activation readouts. AP-1 is an inducible transcriptional complex (classically c-Fos/c-Jun) whose composition and post-translational regulation govern stress-response and differentiation programs, and it is a central mediator of melanoma state plasticity under MAPK inhibition (Bossis et al., 2005; Comandante-Lou et al., 2022). The final variable, p-ERK (Thr202/Tyr204), is a canonical proxy for ERK-pathway activity downstream of MEK and directly reflects perturbation strength and signaling rebound under MAPK inhibition (Roskoski, 2012; Comandante-Lou et al., 2022). Moreover, FRA1 abundance and phosphorylation are tightly coupled to ERK signaling via ERK-dependent stabilization and regulation, making *p-FRA1* a particularly informative bridge between pathway activity and AP-1 state. In our AP1+MAPK matrix, these correspond to the following 0-based column indices: p-cFos(0), p-cJun(1), cFos(4), cJun(6), Fra2(7), JunB(9), JunD(10), Fra1(14), p-Fra1(15), and p-ERK(18).

**Benchmark 2: 4i multiplexed protein imaging (medium-dimensional).** We follow the CellOT benchmark (Bunne et al., 2023) on the 4i multiplexed protein-imaging dataset (Gut et al., 2018), which profiles a 1:1 co-culture of two melanoma tumor cell lines (M130219 and M130429) under DMSO control and a panel of 34 drug treatments, with measurements taken after an 8-hour drug exposure. We report representative results on Imatinib, Trametinib, and Dexamethasone. *Motivation.* This benchmark is valuable precisely because it is *not* a tidy targeted panel: it couples protein intensities with morphology, and it is collected in a co-culture, introducing realistic heterogeneity even within a single "experiment." The three highlighted drugs deliberately span *distinct mechanistic classes* (kinase inhibition vs. steroid receptor signaling), which tests whether conclusions are specific to one biochemical mode of action or persist across qualitatively different response programs. *Data availability:* CellOT benchmark repository and 4i methodology reference.

For the 4i benchmark, we represent each cell by a $d = 48$ feature vector combining protein-intensity summaries with morphology. Specifically, we use (i) 26 compartment-specific intensity features and (ii) 22 geometric descriptors. For protein intensity, we measure 13 markers (CD45, ClCasp3, DAPI, Ki67, MelA, PCNA, Sox9, aTUB, pAKT, pEGFR, pERK, pMET, pS6K1) and, for each marker, include two aggregated compartments: whole-cell and nucleus. Features are named `intensity-cell-<marker>-<stat>` and `intensity-nuclei-<marker>-<stat>`. The aggregation statistic is the mean for all markers except DAPI, for which we use the sum. For morphology, we include the same 11 shape/size de-

scriptors for both the cell and nucleus—area, circularity, convexity, eccentricity, elongation, equivalent_diameter, extent, major_axis_length, mean_radius, perimeter, and roundness—yielding features named `morphology-cell-<descriptor>` and `morphology-nuclei-<descriptor>` (22 total).

**Benchmark 3: SciPlex3 scRNA-seq perturbations (high-dimensional).** We use the SciPlex3 perturbation benchmark (Srivatsan et al., 2020) as a transcriptome-wide setting. *Motivation.* SciPlex3 is the canonical "hard mode" for perturbation prediction: responses live in a high-dimensional gene-expression space where representation learning is often assumed to be essential. Including SciPlex3, therefore guards against a common criticism—that conclusions drawn from protein panels or imaging features may not transfer to transcriptome-scale perturbations. Practically, to keep comparisons meaningful for kernel two-sample evaluation in high dimension, our reporting follows two standard conventions: (i) fit on a larger gene set but compute $\text{MMD}^2$ on highly-variable genes, and (ii) fit and evaluate on the same highly-variable gene subset; within each table, all methods share the same evaluation space and the same bandwidth protocol. *Data availability:* SciPlex3 study page.

For SciPlex3, we report $\text{MMD}^2$ in the highly-variable gene (HVG) space used by the corresponding appendix table. Restricting evaluation to HVGs improves the stability and comparability of kernel two-sample distances in high-dimensional transcriptomic settings, while still capturing population-level transcriptional shifts. We follow the HVG annotation provided with the CellOT benchmark preprocessing for this dataset.

**2i time-course dataset (reprogramming).** We additionally use the reprogramming time-course dataset of (Schiebinger et al., 2019), which provides measurements across multiple time horizons. *Motivation.* Unlike the fixed-readout benchmarks above, this dataset enables a *controlled time-horizon sweep* within a single biological system. That makes it uniquely suited for diagnosing how prediction difficulty changes as the horizon increases—without changing the perturbation, assay, or organismal context. *Data availability:* Cell article page, WOT toolbox, and archived dataset snapshot used in prior work.

For the 2i reprogramming time-course, we report $\text{MMD}^2$ in a fixed highly-variable gene (HVG) subspace, using the top 50 HVGs selected from the preprocessed data. This follows common practice in single-cell time-course benchmarks: restricting to HVGs emphasizes genes that carry the strongest biological variation over time while reducing the influence of near-constant or noise-dominated genes. The resulting medium-dimensional gene space makes kernel two-sample evaluation more stable and comparable across methods, while still capturing the dominant transcriptional shifts induced by the reprogramming dynamics.

## C.2. Evaluation metric: $\text{MMD}^2$ with RBF kernel

We evaluate set-level prediction quality using the squared Maximum Mean Discrepancy ($\text{MMD}^2$), a kernel two-sample distance that is zero if and only if the underlying distributions match (for characteristic kernels) (Gretton et al., 2007; 2012). MMD is well-suited to single-cell perturbation benchmarks because methods typically output *populations* (predicted treated distributions) rather than paired cellwise predictions, and the metric directly quantifies distributional agreement.

Given samples $\mathcal{X} = \{x_i\}_{i=1}^n \subset \mathbb{R}^d$ and $\mathcal{Y} = \{y_j\}_{j=1}^m \subset \mathbb{R}^d$ and a positive definite kernel $k(\cdot, \cdot)$, we use the unbiased empirical estimator

$$\widehat{\text{MMD}}^2(\mathcal{X}, \mathcal{Y}; k) = \frac{1}{n(n-1)} \sum_{i \neq i'} k(x_i, x_{i'}) + \frac{1}{m(m-1)} \sum_{j \neq j'} k(y_j, y_{j'}) - \frac{2}{nm} \sum_{i=1}^n \sum_{j=1}^m k(x_i, y_j). \tag{41}$$

We compute $\widehat{\text{MMD}}^2$ in the same evaluation space used by the corresponding table (e.g., AP-1 JFE features, 4i feature vectors, or an HVG gene subspace).

**RBF kernel.** In all experiments, we use the Gaussian RBF kernel

$$k_\gamma(u, v) = \exp\big(-\gamma \|u - v\|_2^2\big),$$

implemented via standard routines in `sklearn`. The bandwidth parameter $\gamma$ sets the comparison length scale (equivalently, $\gamma = 1/(2\sigma^2)$): larger $\gamma$ emphasizes fine, local discrepancies, while smaller $\gamma$ emphasizes broader distributional shifts.

**Bandwidth selection and leakage control.** Unless stated otherwise, we choose $\gamma$ via the median heuristic *using the training split only* to avoid test leakage. Let $\mathcal{Z}$ denote the pooled training samples used for the MMD comparison (control and treated, restricted to the relevant evaluation space). We set

$$\gamma = \left(\text{median}_{z \neq z' \in \mathcal{Z}} \|z - z'\|_2^2\right)^{-1}.$$

To ensure conclusions are not artifacts of a single bandwidth choice, we additionally report sensitivity results for fixed $\gamma \in \{0.5, 1.0\}$ in the corresponding appendix tables.

MMD with RBF kernels is widely used as a primary distribution-matching metric in single-cell perturbation benchmarks, including (Lotfollahi et al., 2019; Lubeck et al., 2022; Bunne et al., 2023; Chen et al., 2025).

### C.3. Median-heuristic RBF bandwidths across settings

Table 4 reports the median-heuristic bandwidths $\gamma$ for each experimental setting, computed on the corresponding *training split* in the *evaluation space* used for that setting.

*Table 4.* Median-heuristic RBF bandwidths ($\gamma$) used for $\text{MMD}^2$ across experimental settings.

| Dataset / Setting | Split / Condition | Median heuristic $\gamma$ |
|---|---|---|
| 4i (in a drug) | Trametinib | 0.0806 |
| 4i (in a drug) | Imatinib | 0.0574 |
| 4i (in a drug) | Dexamethasone | 0.0744 |
| RNA-seq (SciPlex3) | Trametinib | 0.0022 |
| RNA-seq (SciPlex3) | Givinostat | 0.0022 |
| RNA-seq (SciPlex3) | Abexinostat | 0.0021 |
| AP-1 (replicate) | COLO858 | 0.0487 |
| AP-1 (replicate) | WM902B | 0.0655 |
| AP-1 (replicate) | SKMEL19 | 0.1064 |
| AP-1 (in a cell-line) | COLO858 | 0.0516 |
| AP-1 (in a cell-line) | WM902B | 0.0644 |
| AP-1 (in a cell-line) | SKMEL19 | 0.1048 |
| AP-1 (cross cell-lines) | Train: WM902B, RVH421, SKMEL19 → Test: COLO858 | 0.0655 |
| AP-1 (cross cell-lines) | Train: COLO858, RVH421, SKMEL19 → Test: WM902B | 0.0685 |
| AP-1 (cross cell-lines) | Train: COLO858, WM902B, SKMEL19 → Test: RVH421 | 0.0584 |
| AP-1 (cross cell-lines) | Train: COLO858, WM902B, RVH421 → Test: SKMEL19 | 0.0535 |

# D. Complementary Results

This appendix complements Sections 6 and 5.2 with additional evaluations that probe (i) nonlinear models beyond the trackable regime, (ii) full results for within-context perturbation prediction, (iii) robustness to replicate shifts, (iv) out-of-distribution behavior under cross-cell-line transfer, and (v) a time-course analysis (2i) that varies the prediction horizon. The results are organized as follows: model degradation under coarse temporal measurements (Fig 5), within-context perturbations in 4i (Table 5), within-cell-line AP-1 prediction (Table 6), replicate generalization on AP-1 (Table 7), perturbations on SciPlex3 under two evaluation conventions (Tables 8 and 9), and cross-cell-line AP-1 transfer (Table 10).

### D.1. The challenge of coarse temporal measurements

We examine how time impacts the performance of linear and non-linear models. Using the synthetic setup from Section 5.2.1, we evaluate LAOT alongside the nonlinear baselines on increasing time gaps. Since nonlinear baselines such as scGen and CellOT do not recover an explicit transport map, we use $MMD^2$ as a distribution-level measure between the predicted and ground-truth target populations. Figure 5 shows that all methods achieve low error in the trackable regime, but their errors rise sharply once the time gap enters the NP-hard regime. This degradation is not limited to the linear solver: state-of-the-art nonlinear models exhibit the same collapse, indicating that model expressivity alone does not remove the intrinsic trackability barrier. Thus, the untrackable regime manifests not only as a failure to recover correspondences but also as a failure to predict the target population accurately, even when using nonlinear neural baselines.

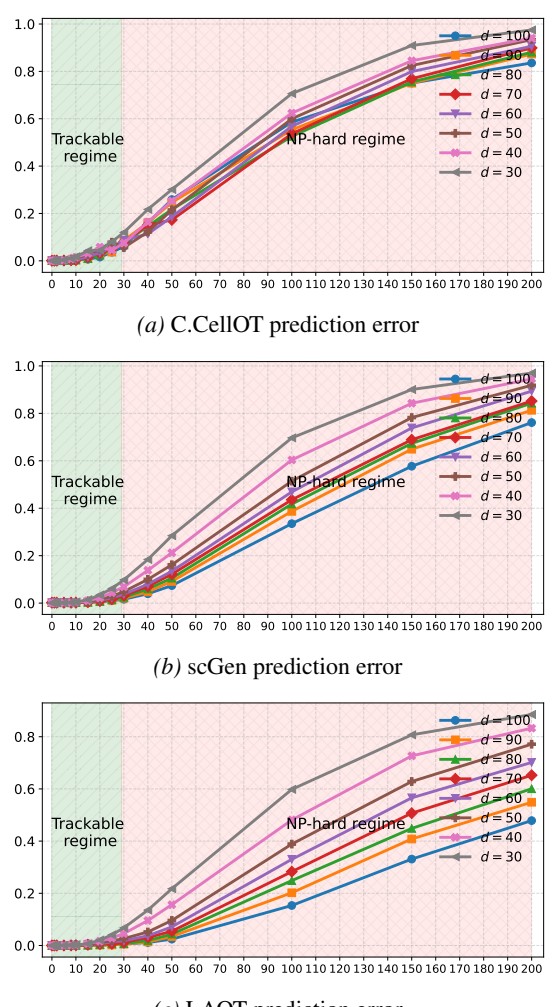

*(a)* C.CellOT prediction error

*(b)* scGen prediction error

*(c)* LAOT prediction error

*Figure 5.* Model degradation in the untrackable regime on synthetic data. We use the same synthetic setup as in Section 5.2.1, but evaluate all methods using $MMD^2$ between predicted and ground-truth target populations. Across LAOT, scGen, and C.CellOT variants, the error remains small in the trackable regime and increases sharply in the NP-hard regime.

## D.2. Within-context benchmarks: 4i and AP-1

Tables 5 and 6 report within-context prediction, where control and treated populations are drawn from the same biological context and differ only through the perturbation protocol. Across the 4i drugs and AP-1 cell lines we consider, the overall pattern is consistent: LAOT ranks highest, whereas Compact_CellOT, CellOT, and scGen are generally less accurate in these settings. Importantly, the advantage of LAOT is not confined to a single perturbation (drug) or a single cell line; it persists across mechanistically distinct 4i drugs and across multiple melanoma lines in AP-1. In addition, the reported std across runs is zero for LAOT, indicating stable performance under repeated trials.

*Table 5.* **4i (mean $\pm$ std).** We report $\text{MMD}^2$ with the median-heuristic bandwidth (computed on the training split), and with fixed $\gamma \in \{0.5, 1.0\}$ for bandwidth sensitivity.

| Drug | Method | $\text{MMD}^2$ ($\gamma = $ median) | $\text{MMD}^2$ ($\gamma = 0.5$) | $\text{MMD}^2$ ($\gamma = 1.0$) |
|---|---|---|---|---|
| **Imatinib** | CellOT | $0.0700 \pm 0.0939$ | $0.0523 \pm 0.0525$ | $0.0247 \pm 0.0233$ |
| | scGen | $0.0330 \pm 0.0034$ | $0.0257 \pm 0.0029$ | $0.0107 \pm 0.0010$ |
| | Compact_CellOT | $0.0079 \pm 0.0035$ | $0.0099 \pm 0.0026$ | $0.0062 \pm 0.0008$ |
| | LAOT | $0.0063 \pm 0.0000$ | $0.0087 \pm 0.0000$ | $0.0050 \pm 0.0000$ |
| **Trametinib** | CellOT | $0.0463 \pm 0.0307$ | $0.0473 \pm 0.0237$ | $0.0268 \pm 0.0131$ |
| | scGen | $0.0098 \pm 0.0020$ | $0.0184 \pm 0.0024$ | $0.0121 \pm 0.0016$ |
| | LAOT | $0.0080 \pm 0.0000$ | $0.0194 \pm 0.0000$ | $0.0153 \pm 0.0000$ |
| | Compact_CellOT | $0.0076 \pm 0.0016$ | $0.0105 \pm 0.0017$ | $0.0076 \pm 0.0008$ |
| **Dexamethasone** | CellOT | $0.0685 \pm 0.0119$ | $0.0608 \pm 0.0114$ | $0.0301 \pm 0.0062$ |
| | scGen | $0.0160 \pm 0.0022$ | $0.0215 \pm 0.0016$ | $0.0113 \pm 0.0008$ |
| | Compact_CellOT | $0.0075 \pm 0.0020$ | $0.0107 \pm 0.0015$ | $0.0072 \pm 0.0006$ |
| | LAOT | $0.0071 \pm 0.0000$ | $0.0132 \pm 0.0000$ | $0.0086 \pm 0.0000$ |

*Table 6.* **AP-1 proteins: cell perturbation (mean $\pm$ std).** We report $\text{MMD}^2$ with the median-heuristic bandwidth (computed on the training split) and with fixed $\gamma \in \{0.5, 1.0\}$.

| Cell line | Method | $\text{MMD}^2$ ($\gamma = $ median) | $\text{MMD}^2$ ($\gamma = 0.5$) | $\text{MMD}^2$ ($\gamma = 1.0$) |
|---|---|---|---|---|
| **COLO858** | CellOT | $0.0995 \pm 0.1007$ | $0.3963 \pm 0.2855$ | $0.4481 \pm 0.2335$ |
| | scGen | $0.0172 \pm 0.0080$ | $0.1096 \pm 0.0199$ | $0.1239 \pm 0.0210$ |
| | Compact_CellOT | $0.0019 \pm 0.0007$ | $0.0220 \pm 0.0106$ | $0.0301 \pm 0.0142$ |
| | LAOT | $0.0006 \pm 0.0000$ | $0.0150 \pm 0.0000$ | $0.0203 \pm 0.0000$ |
| **WM902B** | CellOT | $0.0443 \pm 0.0394$ | $0.1692 \pm 0.1104$ | $0.2175 \pm 0.1323$ |
| | scGen | $0.1423 \pm 0.0293$ | $0.2664 \pm 0.0539$ | $0.2258 \pm 0.0468$ |
| | Compact_CellOT | $0.0015 \pm 0.0006$ | $0.0052 \pm 0.0010$ | $0.0061 \pm 0.0008$ |
| | LAOT | $0.0007 \pm 0.0000$ | $0.0096 \pm 0.0000$ | $0.0130 \pm 0.0000$ |
| **SKMEL19** | CellOT | $0.1122 \pm 0.0855$ | $0.2498 \pm 0.1509$ | $0.3062 \pm 0.1303$ |
| | scGen | $0.0323 \pm 0.0218$ | $0.0697 \pm 0.0250$ | $0.0688 \pm 0.0164$ |
| | Compact_CellOT | $0.0016 \pm 0.0004$ | $0.0057 \pm 0.0011$ | $0.0073 \pm 0.0012$ |
| | LAOT | $0.0011 \pm 0.0000$ | $0.0061 \pm 0.0000$ | $0.0079 \pm 0.0000$ |

## D.3. Replication generalization on AP-1

Table 7 evaluates cross-replicate generalization, which stress-tests robustness to measurement noise, batch effects, and replicate-specific artifacts while keeping the biological context fixed. The main qualitative conclusion is that the two correspondence-aware baselines (LAOT and Compact_CellOT) remain the most reliable performers under replicate shift.

*Table 7.* **AP-1 proteins: Replicate experiment (mean $\pm$ std).** We report MMD$^2$ with the median-heuristic bandwidth (computed on the training split) and with fixed $\gamma \in \{0.5, 1.0\}$ for bandwidth sensitivity.

| Cell line | Method | MMD$^2$ ($\gamma$ = median) | MMD$^2$ ($\gamma$ = 0.5) | MMD$^2$ ($\gamma$ = 1.0) |
|---|---|---|---|---|
| **COLO858** | CellOT | 0.1109$\pm$ 0.1068 | 0.3582$\pm$ 0.1757 | 0.3920$\pm$ 0.1682 |
| | scGen | 0.0779$\pm$ 0.0206 | 0.2788$\pm$ 0.0428 | 0.2919$\pm$ 0.0371 |
| | Compact_CellOT | 0.0116$\pm$ 0.0020 | 0.0485$\pm$ 0.0094 | 0.0525$\pm$ 0.0149 |
| | LAOT | 0.0069$\pm$ 0.0000 | 0.0468$\pm$ 0.0000 | 0.0525$\pm$ 0.0000 |
| **WM902B** | CellOT | 0.0715$\pm$ 0.0480 | 0.2036$\pm$ 0.1411 | 0.2290$\pm$ 0.1532 |
| | scGen | 0.1376$\pm$ 0.0384 | 0.3013$\pm$ 0.0726 | 0.2685$\pm$ 0.0581 |
| | Compact_CellOT | 0.0095$\pm$ 0.0022 | 0.0201$\pm$ 0.0029 | 0.0189$\pm$ 0.0026 |
| | LAOT | 0.0085$\pm$ 0.0000 | 0.0246$\pm$ 0.0000 | 0.0270$\pm$ 0.0000 |
| **SKMEL19** | CellOT | 0.0849$\pm$ 0.0697 | 0.1868$\pm$ 0.1043 | 0.2331$\pm$ 0.0661 |
| | scGen | 0.0323$\pm$ 0.0218 | 0.0697$\pm$ 0.0250 | 0.0688$\pm$ 0.0164 |
| | LAOT | 0.0138$\pm$ 0.0000 | 0.0263$\pm$ 0.0000 | 0.0232$\pm$ 0.0000 |
| | Compact_CellOT | 0.0097$\pm$ 0.0028 | 0.0178$\pm$ 0.0044 | 0.0168$\pm$ 0.0035 |

## D.4. Transcriptomic perturbations: SciPlex3 under two evaluation conventions

Tables 8 and 9 report SciPlex3 results under two standard conventions: (i) fitting on a larger gene set while evaluating in a restricted highly-variable gene space, and (ii) fitting and evaluating within the same restricted gene space. Following the original CellOT protocol for high-dimensional transcriptomic data, CellOT uses scGen as a dimensionality-reduction component before learning the transport map. Considering both evaluation conventions is useful because it separates two effects: the modeling choice, namely how much transcriptomic information is used during fitting, and the evaluation choice, namely the space in which distributional agreement is assessed. Across drugs and across both conventions, the qualitative ranking is stable: LAOT is consistently competitive and typically among the strongest methods, with scGen and CellOT trailing behind depending on the drug and the evaluation convention.

*Table 8.* **RNA-seq: SciPlex3 (mean $\pm$ std).** Models are fit on 1,000 genes, while evaluation is computed using only the top 100 genes. We report MMD$^2$ with the median-heuristic bandwidth (computed on the training split) and with fixed $\gamma \in \{0.5, 1.0\}$ for bandwidth sensitivity.

| Drug | Method | MMD$^2$ ($\gamma$ = median) | MMD$^2$ ($\gamma$ = 0.5) | MMD$^2$ ($\gamma$ = 1.0) |
|---|---|---|---|---|
| **Trametinib** | CellOT | 0.0078$\pm$ 0.0020 | 0.0037$\pm$ 0.0002 | 0.0032$\pm$ 0.0000 |
| | scGen | 0.0059$\pm$ 0.0014 | 0.0016$\pm$ 0.0003 | 0.0003$\pm$ 0.0000 |
| | Compact_CellOT | 0.0048$\pm$ 0.0010 | 0.0038$\pm$ 0.0001 | 0.0033$\pm$ 0.0000 |
| | LAOT | 0.0040$\pm$ 0.0000 | 0.0002$\pm$ 0.0000 | 0.0001$\pm$ 0.0000 |
| **Givinostat** | CellOT | 0.0117$\pm$ 0.0029 | 0.0031$\pm$ 0.0002 | 0.0029$\pm$ 0.0000 |
| | scGen | 0.0083$\pm$ 0.0007 | 0.0017$\pm$ 0.0003 | 0.0002$\pm$ 0.0000 |
| | Compact_CellOT | 0.0079$\pm$ 0.0011 | 0.0031$\pm$ 0.0001 | 0.0029$\pm$ 0.0000 |
| | LAOT | 0.0033$\pm$ 0.0000 | 0.0001$\pm$ 0.0000 | 0.0000$\pm$ 0.0000 |
| **Abexinostat** | CellOT | 0.0129$\pm$ 0.0063 | 0.0022$\pm$ 0.0000 | 0.0022$\pm$ 0.0000 |
| | scGen | 0.0091$\pm$ 0.0012 | 0.0002$\pm$ 0.0000 | 0.0000$\pm$ 0.0000 |
| | Compact_CellOT | 0.0074$\pm$ 0.0019 | 0.0022$\pm$ 0.0000 | 0.0022$\pm$ 0.0000 |
| | LAOT | 0.0038$\pm$ 0.0000 | 0.0000$\pm$ 0.0000 | 0.0000$\pm$ 0.0000 |

*Table 9.* **RNA-seq: SciPlex3 (mean $\pm$ std).** Models are fit on the top 100 genes; results are computed using only the top 100 genes. We report MMD$^2$ with the median-heuristic bandwidth (computed on the training split) and with fixed $\gamma \in \{0.5, 1.0\}$ for bandwidth sensitivity.

| Drug | Method | MMD$^2$ ($\gamma$ = median) | MMD$^2$ ($\gamma$ = 0.5) | MMD$^2$ ($\gamma$ = 1.0) |
|---|---|---|---|---|
| **Trametinib** | CellOT | 0.0050$\pm$ 0.0009 | 0.0034$\pm$ 0.0000 | 0.0033$\pm$ 0.0000 |
| | scGen | 0.0049$\pm$ 0.0011 | 0.0002$\pm$ 0.0000 | 0.0001$\pm$ 0.0000 |
| | Compact_CellOT | 0.0040$\pm$ 0.0006 | 0.0033$\pm$ 0.0000 | 0.0032$\pm$ 0.0000 |
| | LAOT | 0.0022$\pm$ 0.0000 | 0.0007$\pm$ 0.0000 | 0.0002$\pm$ 0.0000 |
| **Givinostat** | CellOT | 0.0104$\pm$ 0.0017 | 0.0041$\pm$ 0.0002 | 0.0032$\pm$ 0.0001 |
| | scGen | 0.0055$\pm$ 0.0010 | 0.0004$\pm$ 0.0000 | 0.0002$\pm$ 0.0000 |
| | Compact_CellOT | 0.0057$\pm$ 0.0018 | 0.0030$\pm$ 0.0000 | 0.0029$\pm$ 0.0000 |
| | LAOT | 0.0025$\pm$ 0.0000 | 0.0011$\pm$ 0.0000 | 0.0003$\pm$ 0.0000 |
| **Abexinostat** | CellOT | 0.0100$\pm$ 0.0027 | 0.0023$\pm$ 0.0002 | 0.0023$\pm$ 0.0000 |
| | scGen | 0.0042$\pm$ 0.0006 | 0.0000$\pm$ 0.0000 | 0.0000$\pm$ 0.0000 |
| | Compact_CellOT | 0.0059$\pm$ 0.0018 | 0.0023$\pm$ 0.0000 | 0.0022$\pm$ 0.0000 |
| | LAOT | 0.0019$\pm$ 0.0000 | 0.0000$\pm$ 0.0000 | 0.0000$\pm$ 0.0000 |

## D.5. Cross-cell-line generalization (out-of-distribution)

Table 10 reports cross-cell-line transfer, where models are trained on several cell lines and evaluated on a held-out line. This setting is qualitatively harder than within-cell-line evaluation, and the table reflects that difficulty: all methods exhibit a pronounced degradation relative to within-context results, and performance becomes more heterogeneous across target cell lines. No single method dominates uniformly across all held-out lines; rather, which method is best can depend on the target context. This behavior is consistent with the view that cross-cell-line transfer introduces additional biological shift beyond the within-context setting and therefore represents a substantially different generalization regime.

*Table 10.* **AP-1 proteins: cross cell-line perturbation (mean $\pm$ std).** We report MMD$^2$ with the median-heuristic bandwidth (computed on the training split) and with fixed $\gamma \in \{0.5, 1.0\}$ for bandwidth sensitivity. For each column, models are trained on the remaining three cell lines and evaluated on the held-out cell line under the same perturbation protocol.

| Held-out cell line | Method | MMD$^2$ ($\gamma$ = median) | MMD$^2$ ($\gamma$ = 0.5) | MMD$^2$ ($\gamma$ = 1.0) |
|---|---|---|---|---|
| **COLO858** | Compact_CellOT | 0.2269$\pm$ 0.0270 | 0.3202$\pm$ 0.0117 | 0.1884$\pm$ 0.0099 |
| | CellOT | 0.1581$\pm$ 0.0554 | 0.4350$\pm$ 0.1312 | 0.4497$\pm$ 0.1210 |
| | scGen | 0.1022$\pm$ 0.0402 | 0.2182$\pm$ 0.0252 | 0.1900$\pm$ 0.0139 |
| | LAOT | 0.0666$\pm$ 0.0000 | 0.1964$\pm$ 0.0000 | 0.1518$\pm$ 0.0000 |
| **WM902B** | scGen | 0.3105$\pm$ 0.0695 | 0.4009$\pm$ 0.0544 | 0.2970$\pm$ 0.0345 |
| | CellOT | 0.1721$\pm$ 0.0163 | 0.6432$\pm$ 0.0568 | 0.8174$\pm$ 0.0701 |
| | LAOT | 0.1709$\pm$ 0.0000 | 0.2841$\pm$ 0.0000 | 0.2002$\pm$ 0.0000 |
| | Compact_CellOT | 0.1197$\pm$ 0.0249 | 0.1913$\pm$ 0.0313 | 0.1456$\pm$ 0.0314 |
| **RVH421** | CellOT | 0.0994$\pm$ 0.0404 | 0.3697$\pm$ 0.0880 | 0.4739$\pm$ 0.1308 |
| | Compact_CellOT | 0.0990$\pm$ 0.0054 | 0.2276$\pm$ 0.0131 | 0.1784$\pm$ 0.0136 |
| | LAOT | 0.0987$\pm$ 0.0000 | 0.2065$\pm$ 0.0000 | 0.1632$\pm$ 0.0000 |
| | scGen | 0.0966$\pm$ 0.0246 | 0.2570$\pm$ 0.0442 | 0.2276$\pm$ 0.0306 |
| **SKMEL19** | LAOT | 0.2542$\pm$ 0.0000 | 0.2848$\pm$ 0.0000 | 0.1658$\pm$ 0.0000 |
| | CellOT | 0.2286$\pm$ 0.1592 | 0.2472$\pm$ 0.0560 | 0.2131$\pm$ 0.0344 |
| | Compact_CellOT | 0.2174$\pm$ 0.0348 | 0.2450$\pm$ 0.0150 | 0.1405$\pm$ 0.0063 |
| | scGen | 0.1174$\pm$ 0.0162 | 0.2912$\pm$ 0.0305 | 0.2423$\pm$ 0.0297 |

### D.6. Time-course study (2i condition): error versus prediction horizon

Figure 6 reports a horizon-sweep on the 2i time-course dataset: we fix the pre-state at the time of drug addition and vary the post time point along the trajectory. The qualitative behavior is clear and consistent: distributional prediction error increases with the pre/post time gap, and the smallest errors occur at the shortest horizons. This monotone "horizon effect" holds across kernel bandwidths (Panels 6a–6c), indicating that it is not an artifact of a particular RBF length scale but a robust property of the task along the time-course.

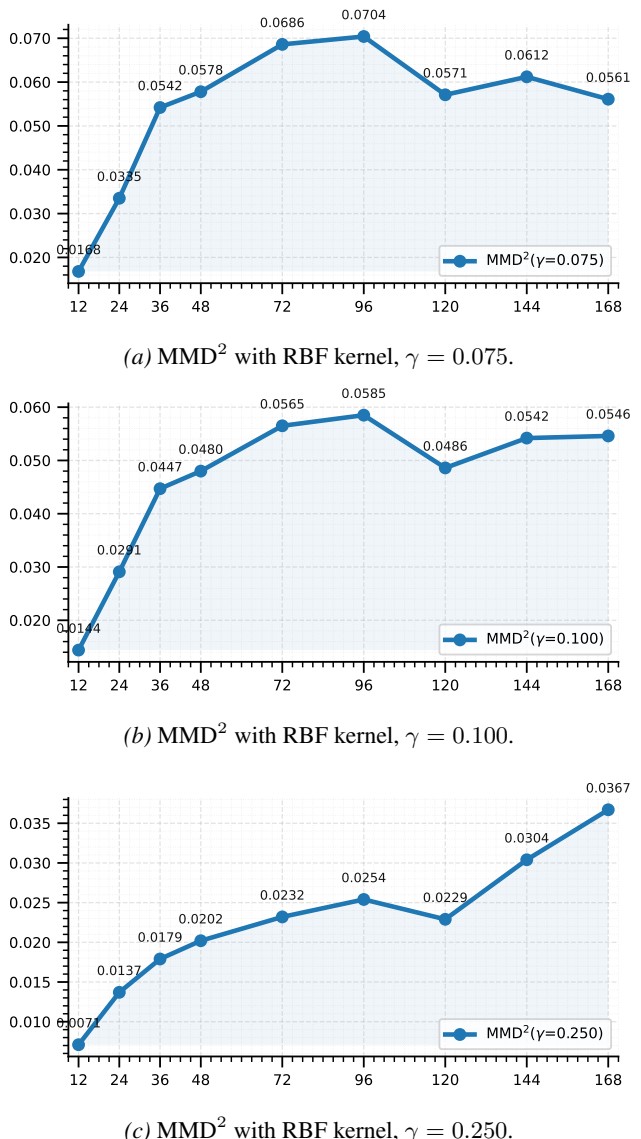

*(a)* $\mathrm{MMD}^2$ with RBF kernel, $\gamma = 0.075$.

*(b)* $\mathrm{MMD}^2$ with RBF kernel, $\gamma = 0.100$.

*(c)* $\mathrm{MMD}^2$ with RBF kernel, $\gamma = 0.250$.

*Figure 6.* **Forecasting error across time horizons in the 2i dataset.** We analyze the 2i time-course dataset, sampled every 12 hours. For each prediction horizon $\Delta t \in \{12, 24, \ldots, 168\}$ hours, we learn a pre→post mapping on training cells and evaluate on $n = 500$ held-out test cells. Each curve reports the test $\mathrm{MMD}^2$ between the predicted post-state distribution and the true post-state distribution. Consistently across kernel bandwidths $\gamma$, shorter horizons (12–24h) yield smaller $\mathrm{MMD}^2$, while larger horizons exhibit increased discrepancy, indicating that distributional prediction becomes harder as the temporal gap grows.

## Summary of Changes After the Rebuttal

We sincerely thank the reviewers for their thoughtful and constructive feedback. After the rebuttal period, we revised the manuscript to address the reviewers' concerns, clarify the scope of our claims, and strengthen the empirical evidence supporting our main position.

- **Added Appendix D.1 on the untrackable regime.** We added new synthetic experiments in Section D.1 showing that LAOT, scGen, and Compact CellOT degrade as the measurement gap moves into the untrackable/NP-hard regime. These results strengthen the empirical support for the paper's central regime-dependent claim.

- **Clarified the meaning of correspondence in Section 3.** We clarified that true cell-level matching between pre- and post-perturbation states is not directly observable in biological assays because single-cell measurements are destructive. Therefore, the matching in our formulation represents a latent correspondence between two observed populations, not direct tracking of the same physical cells.

- **Revised the discussion of cross-cell-line generalization in Section 8.** We revised the cross-cell-line transfer discussion to connect its empirical degradation to violations of the within-context and smooth-transition assumptions. This clarifies that cross-context prediction represents a harder regime in which no evaluated method is consistently reliable.

- **Expanded the discussion and limitations in Section 8.** We clarified that our claim is regime-dependent. We do not argue that biological perturbations are universally linear; rather, we show that LAOT can suffice when the experimental design keeps the problem in the trackable regime.

- **Strengthened the baseline comparisons in subsection 6.2.** We added Compact CellOT to the main SciPlex3 scRNA-seq benchmark table in Subsection 6.2. This provides a more complete comparison with nonlinear baselines and helps separate the effect of model capacity from the default CellOT architecture.

- **Added a limitation on balanced transport in Section 8.** We clarified that our theoretical formulation uses a balanced transport abstraction based on permutation matching. Since real perturbations may involve cell proliferation, cell death, or changes in population mass, we identify extensions to unbalanced or stochastic optimal transport as an important future direction.

