# OpenReview forum: "Position: Temporal Measurement Interval Determines Computational and Model Complexity in Single-Cell Perturbation Analysis"
_ICML.cc/2026/Position_Paper_Track — ICML 2026 Position Paper Track spotlight_

### Official Review · Reviewer_s82B · 2026-03-12

**Significance:** 4
**Argument Clarity:** 3
**Rating:** 5
**Confidence:** 4

**Questions:**

See weaknesses above.

**Alternative Views Section:**

Yes

**Compliance With Llm Reviewing Policy A Conservative:**

Affirmed.

**Discussion Potential:**

3

**Paper Summary:**

This paper introduces an interesting claim that the measurement time gap is the key experimental knob controlling both the computational tractability of coupling and the effective model complexity, instead of simply deepening the complexity of neural network. Theory 4.5 is given to support the claim. Extensive experiments on synthetic and real-world experiments are provided.

**Position:**

Yes

**Position In Title:**

Yes

**Related Work:**

3

**Strengths And Weaknesses:**

Strengths:
- The paper is clearly-written;
- The focused task is important;
- The proposed position is novel and interesting, also reasonable;
- Theoretical and empirical evidences are given to support the claim.

Weaknesses:
- In table 3, the performance is not very well in the held-out cell line, you may provide more explanations.

**Support:**

4

---

> ### Author Rebuttal · Authors · 2026-03-30
>
> **Response to Reviewer s82B**
>
> We thank the reviewer for the positive assessment and for pointing out the cross-cell-line result as an important place to clarify the paper.
>
> We agree that Table 3 requires more explanation. The held-out-cell-line setting is intentionally much harder than the within-context setting because the biological context itself changes. That change alters both the geometry of the pre-perturbation population and the form of the perturbation response, so the task is no longer simply to recover a coupling and estimate one transition map within a fixed context. Instead, it combines coupling recovery with a substantial out-of-distribution shift.
> This is also a setting in which the assumptions of our theory may no longer hold. In particular, when the source and target populations come from different underlying cell distributions, the problem goes beyond the regime addressed by our current analysis. We therefore view this as an open problem and an important direction for future work, especially in settings where the distributions of cells differ significantly across contexts. We use this experiment to show scope of our theoretical results and elaborate that a linear model is not always a good choice.
>
> As suggested by the reviewer, we will expand the discussion around Table 3 to make it clearer.

---

> > ### Author Rebuttal · Reviewer_s82B · 2026-04-02
> >
> > Thanks for the response, which addressed my question. Considering that i have already given a rather high score, i would like to maintain my positive score 5.

---

### Official Review · Reviewer_Y6ic · 2026-03-13

**Significance:** 2
**Argument Clarity:** 3
**Rating:** 5
**Confidence:** 3

**Questions:**

1. While the Lipschitz constant $L$ is unknown, are there biological heuristics or prior literature on protein/RNA half-lives that could help researchers roughly bound $L$ to estimate $\Delta$ before running a pilot time-course experiment?
2. The theory perfectly addresses linear transitions near the identity matrix. Are there counter-examples, eg, rapid, highly non-linear biological shock responses (e.g., immediate stress responses under 1 hour) that would violate the linear approximation even if the time gap strictly satisfies $t < \Delta$?
3. How does the framework account for cellular proliferation or death (unbalanced optimal transport) occurring within the measurement interval, which might violate the strict permutation matching assumption?

**Alternative Views Section:**

Yes

**Compliance With Llm Reviewing Policy A Conservative:**

Affirmed.

**Discussion Potential:**

3

**Final Justification:**

My initial rating was already good, rebuttal is only for resolving minor issues; No change in score.

**Paper Summary:**

This position paper challenges the prevailing trend in single-cell perturbation analysis, which typically relies on highly expressive, non-linear neural networks (like CellOT or scGen) to map unpaired pre- and post-perturbation cellular states. The authors propose that the measurement time gap between observations is the fundamental factor determining both computational tractability and required model complexity. They theoretically establish a "Nyquist-like" critical time gap, $\Delta$. Below $\Delta$, matching cells is polynomial-time solvable and transitions can be modeled linearly; above $\Delta$, recovering the matching becomes NP-hard in the worst case, even for linear transitions. The theory relies on the Lipschitz continuity of the transition dynamics and the Restricted Isometry Property (RIP) of the initial states. The authors empirically validate this phase transition using a minimal linear solver, LAOT, across synthetic data and multiple real-world biological datasets (AP-1, 4i, SciPlex3, and a 2i time-course), demonstrating that linear models match or exceed state-of-the-art non-linear methods in the short-gap regime. Ultimately, they advocate for a "design-first" paradigm where experimental timing dictates the analytical approach.

**Position:**

Yes

**Position In Title:**

Yes

**Related Work:**

4

**Strengths And Weaknesses:**

#### Strengths

* The paper offers a highly original perspective by reframing a modeling bottleneck (complex neural transport maps) as an experimental design issue (sampling rate). This challenges the community to rethink post-hoc model selection.


* The authors provide a solid mathematical justification for their claims (Theorem 4.5), which defines the precise bound for $\Delta$ and formally bridges the problem to NP-hardness.


* The claims are not left as purely theoretical. The authors benchmark across complementary, standard datasets spanning targeted protein panels, multiplexed imaging, and high-dimensional scRNA-seq. The sharp performance drop-off demonstrated in the 2i time-course data beautifully mirrors their theoretical phase transition.


* By introducing LAOT, the authors successfully prove that a zero-parameter linear map can outperform heavily parameterized architectures like CellOT when the temporal assumptions hold, highlighting the inefficiency of current baseline methods in trackable regimes. This is a strong evidence for their position.



#### Weaknesses

* As the authors acknowledge, verifying the RIP assumption for real biological data matrices is generally hard and impractical. This makes it difficult to definitively prove the dataset resides in the trackable regime a priori without running pilot tests.


* The critical threshold $\Delta$ depends on the Lipschitz constant $L$, which is unknown before-hand. Hence, biologists must still rely on empirical scans to find the "operationally tractable window".


* The linear assumption collapses under significant distribution shifts, such as cross-cell-line generalization. While the authors transparently report this failure (and note baselines also fail), it limits the scope of the linear approach solely to highly controlled, within-context experimental setups. Does this mean the position may still be invalid under the other conditions? But even so, the advocation for this domain is a contribution and a valid position. I just would appreciate a bit more discussion on this limitation.

**Support:**

4

---

> ### Author Rebuttal · Authors · 2026-03-30
>
> **Response to Reviewer Y6ic**
>
> We thank the reviewer for the careful reading and for identifying the design-first perspective as the main conceptual contribution of the paper.
>
>
> > “RIP is hard to verify and L is unknown a priori. Could biological heuristics help roughly bound L?”
>
> **Respond:** As noted in the discussion, these constants are not known a priori. In practice, our lightweight model, LAOT, can be used to estimate the $\Delta$ value through a short pilot time-course. For example, in Figure 3, LAOT exhibits a phase transition at roughly 36 hours, suggesting that time gaps beyond 36 hours may introduce substantially greater complexity in this 2i perturbation setting. We will clarify and expand this point in the camera-ready version.
> We will revise the discussion to make this point clearer: estimating the tractable window is still primarily an empirical question. We view the development of sharper biologically grounded estimates, such as protein/RNA half-lives, as an important direction for future work, likely requiring closer collaboration with domain experts.
>
> > “How should the position be interpreted under cross-cell-line generalization?”
>
>
> **Respond:**  Our position is developed for a specific problem setting: when the assumptions and conditions of our analysis do not hold, our theorems cannot be applied directly. This negative result highlights the scope of our result. More broadly, we view the difficulty of cross-cell-line generalization as an important open problem that requires significant future work, especially since none of the existing baselines achieves consistently strong performance in this setting. We will clarify and expand on this point in the camera-ready version of the paper.”
>
> > “Does cross-cell-line failure restrict the scope of the linear approach?"
>
> **Respond:**  Yes, this failure highlights a limitation of the linear method and shows that a linear model is not always the most appropriate choice. Moreover, this experiment indicates that existing baselines also do not necessarily perform well in this regime.
>
> > “Are there short-time but highly nonlinear counterexamples?”
>
> **Respond:** Absolutely. The main theorem (4.5) does not require the model to be linear. In part a (trackable regime), the problem casts to a potentially non-linear supervised learning. We use a linear model to elaborate on the simplicity of current short-time benchmarks on which very complex models are often used.  Indeed, we are working on a follow up research problem which indeed requires using a non-linear supervised learning after finding the matching.
>
> > “How does the framework handle proliferation or death / unbalanced OT?”
>
> **Respond:** In real biological perturbation assays, “correct recovery” should not be interpreted as recovering a literal biological identity matching. Rather, it should be interpreted as recovering a latent transport from one distribution to another. The seminal Brenier’s theorem proves this transport exists under weak assumptions [Brenier, Polar Factorization, CPAM 1991]. The OT-based cell perturbation analysis is built precisely on finding this transport map using a min-max optimization [Bunne et al., Neural Optimal Transport for Single-Cell Perturbation, Nature Methods23].
>
> Algorithm 1 (LAOT) is intentionally designed in the balanced, permutation-based setting. We believe that extending the time-dependent complexity analysis to unbalanced transport is an important next step. We will discuss this in future revisions. Thanks for your constructive comment.

---

> > ### Author Rebuttal · Reviewer_Y6ic · 2026-04-04
> >
> > Nice, pls revise accordingly and I am looking forward to your polished version.
> >
> > My score was already high, so I maintain.

---

### Official Review · Reviewer_9cM9 · 2026-03-16

**Significance:** 3
**Argument Clarity:** 4
**Rating:** 5
**Confidence:** 4

**Questions:**

- If LAOT is run on bootstrap resamples of the same dataset, does it recover consistent permutation matrices?
- Can the authors include some of the baselines in the synthetic data analysis?

**Alternative Views Section:**

Yes

**Compliance With Llm Reviewing Policy A Conservative:**

Affirmed.

**Discussion Potential:**

3

**Final Justification:**

This is a strong work with clear contributions. Rebuttal addressed my few remaining questions

**Paper Summary:**

The authors argue that the time gap between pro and perturbation in perturb-seq experiments is an important unmodeled covariate in the current set of models. The author has decomposed the problem into two components: 1 learning a transition map $F_t$ recovering a permutation matrix $\Pi$ that matches pre and post perturbation cells. In Theorem 4.5, they prove that a phase transition below the time gap is recoverable in polynomial time, and above it the problem is NP-hard for linear maps. They go on to validate their findings in a synthetic data setting and three biological benchmarks. Their linear solver LAOT matches or outperforms cellOT and other benchmarks on tasks with a short measurement gap.

**Position:**

Yes

**Position In Title:**

Yes

**Related Work:**

3

**Strengths And Weaknesses:**

### Strengths
- The paper contains a nice theoretical contribution analyzing the cell perturbation problem through the delta time gap, providing theoretical tools for identifying it and validating it with synthetic data.
- The position paper is consistent with other findings that linear models perform comparably to deep learning tools on perturbation prediction. This paper can offer an explanation grounded in this temporal analysis.
- The paper is clearly written

### Weaknesses
- Compact CellOT performed comparably to LAOT on a number of benchmarks. This can weaken the claim that recovering the pre- and post-perturbation samples is what drives performance. Is it possible that it's just low-capacity model capacity sufficient in this short-gap regime?
- Compact CellOT is missing from the SciPlex3 RNA-seq results
- Given that perturbability is destructive, there is no ground truth pairing between pre and post-perturbation cells. How does the learned permutation approximation matching? The paper does not discuss what the correct recovery means in the setting.
- There's no test whether the learned perturbation is stable across the splits. I'd be interested in seeing whether LAoT under the cross-validation setting is consistent and is able to recover the matching.

**Support:**

4

---

> ### Author Rebuttal · Authors · 2026-03-30
>
> **Response to Reviewer 9cM9**
>
> We thank the reviewer for the thoughtful comments and for highlighting both our theoretical contribution on temporal phase transition and the broader relevance of competitive linear models in perturbation prediction.
>
> > “There is no ground-truth pairing in destructive perturbation assays; what does correct recovery mean?
>
> **Respond:** In real biological perturbation assays, the measurements are destructive, so there is no observable cell-identity-level pairing between pre- and post-perturbation populations. In this setting, “correct recovery” should not be interpreted as recovering a literal biological identity matching. Rather, it should be interpreted as recovering a latent transport from one distribution to another. The seminal Brenier’s theorem proves this transport exists under weak assumptions [Brenier, Polar Factorization, CPAM 1991]. The OT-based cell perturbation analysis is built precisely on finding this transport map using a min-max optimization [Bunne et al., Neural Optimal Transport for Single-Cell Perturbation, Nature Methods23]. We will discuss this in future revisions. Thanks for your constructive comment.
>
> > “Compact CellOT performed comparably to LAOT on some benchmarks; perhaps low capacity alone is sufficient.”
>
> **Respond:** Reducing the model capacity alone may not be a solution. We observed that the main challenge of CellOT is min-max optimization, which is known to be particularly more challenging than minimization problems when the function is not convex. For example, [Adolphs et al., Local Saddle Point Optimization AISTATS19] shows that min-max optimization with gradient optimization admits stable attractors that are not locally optimum, which is in clear contrast to minimization problems. We illustrated this unstable optimization in Figure 4 for both CellOT and Compact CellOT. But why is Compact CellOT performing well on some benchmarks? We believe that this performance is due to the fact that these benchmarks are in a favourable regime, when even a linear model (LAOT) performs well.
>
> > “Can LAOT be tested for consistency across splits or bootstrap resamples?”
>
> **Respond:** We thank the reviewer for this helpful suggestion. We agree that a robustness analysis would strengthen the paper, and in the revision, we plan to add results comparing LAOT and the baselines across different data splits. Due to page limits, we placed the replicate generalization experiments in Appendix E.2. In the camera-ready version, we may move some of these results into the main text or add a cross-validation experiment.
>
> Bootstrap resampling is less suitable in our setting because repeated and omitted observations can violate the RIP assumption and make permutation comparisons less well defined. For this reason, we believe evaluation across multiple data splits is a more appropriate way to assess robustness.
>
> > “Can the authors include baselines in the synthetic analysis?”
>
> **Respond:** Yes, we have prepared a few plots of baseline results for the synthetic analysis. We will include them in the camera-ready version. Here, we can provide tables only. Below, we report two tables summarizing the behavior of the baselines on the synthetic data; additional results will be included in the final paper. Since the baselines do not output permutation matrices, we evaluate them using the MMD error of their predictions. As the tables show, the error is close to zero when the controlled time $t$ is short, but it increases as the time becomes larger. Thank you for this constructive suggestion.
>
> Dimension 50
> |Method/time|10|15|20|25|30|40|50|100|150|200|
> |---:|---:|---:|---:|---:|---:|---:|---:|---:|---:|---:|
> |scGen|0.00|0.01|0.02|0.03|0.05|0.10|0.16|0.51|0.78|0.92|
> |CellOT|0.01|0.01|0.04|0.06|0.10|0.15|0.25|0.61|0.84|0.94|
> |LAOT|0.00|0.00|0.01|0.02|0.03|0.05|0.10|0.39|0.63|0.77|
>
> Dimension 100
> |Method/time|10|15|20|25|30|40|50|100|150|200|
> |---|---:|---:|---:|---:|---:|---:|---:|---:|---:|---:|
> |scGen |0.00|0.00|0.01|0.01|0.02|0.04|0.07|0.33|0.58|0.76|
> |CellOT|0.00|0.00|0.01|0.01|0.02|0.13|0.20|0.54|0.71|0.86|
> |LAOT|0.00|0.00|0.00|0.00|0.01|0.01|0.02|0.15|0.33|0.48|
>
> > “Compact CellOT is missing from the SciPlex3 RNA-seq results.”
>
> **Respond:**  We agree. We have now added the SciPlex3 RNA-seq results for Compact CellOT. Thanks for your constructive comment.
>
> Completed version of Table 2
> |Method|Trametinib|Givinostat|Abexinostat|
> |---|---|---|---|
> |CellOT|0.0078±0.0020|0.0117±0.0029|0.0129±0.0063|
> |scGen|0.0059±0.0014|0.0083±0.0007|0.0091±0.0012|
> |CompactCellOT|0.0048±0.0010|0.0079±0.0011|0.0074±0.0019|
> |LAOT|**0.0040±0.0000**|**0.0033±0.0000**|**0.0038±0.0000**|
>
> Completed version of Table 9
> |Method|Trametinib|Givinostat|Abexinostat|
> |---|---|---|---|
> |CellOT|0.0050±0.0009|0.0104±0.0017|0.0100±0.0027|
> |scGen|0.0049±0.0011|0.0055±0.00010|0.0042±0.0006|
> |CompactCellOT|0.0040±0.0006|0.0057±0.0018|0.0059±0.0018|
> |LAOT|**0.0022±0.0000**|**0.0025±0.0000**|**0.0019±0.0000**|

---

> > ### Author Rebuttal · Reviewer_9cM9 · 2026-04-02
> >
> > Great work thank you

---

### Decision · Program_Chairs · 2026-04-30

**Decision:**

Accept (spotlight)

**Comment:**

This position paper argues that the temporal measurement interval fundamentally determines computational tractability and model complexity in single-cell perturbation analysis. Reviewers consistently found the paper clear, novel, and well-supported both theoretically and empirically, and noted its potential to stimulate discussion by reframing a modeling challenge as an experimental design issue. The concerns raised in the initial reviews were addressed in the rebuttal and the responses were appreciated by the reviewers. Given the strong consensus and the paper’s clear and timely position, I recommend acceptance.